# ENTROPY IS NOT ENOUGH FOR TEST-TIME ADAPTATION: FROM THE PERSPECTIVE OF DISENTANGLED FACTORS

**Jonghyun Lee**[1*]    **Dahuin Jung**[2*]    **Saehyung Lee**[1]    **Junsung Park**[1]
**Juhyeon Shin**[3]    **Uiwon Hwang**[4†]    **Sungroh Yoon**[1,3,5†]
[1]Department of Electrical and Computer Engineering, Seoul National University
[2]School of Computer Science and Engineering, Soongsil University
[3]Interdisciplinary Program in Artificial Intelligence, Seoul National University
[4]Division of Digital Healthcare, Yonsei University
[5]AIIS, ASRI, INMC, and ISRC, Seoul National University
`leejh9611@snu.ac.kr, dahuin.jung@ssu.ac.kr, halo8218@snu.ac.kr,`
`{jerryray, newjh12}@snu.ac.kr, uiwon.hwang@yonsei.ac.kr, sryoon@snu.ac.kr`

## ABSTRACT

Test-time adaptation (TTA) fine-tunes pre-trained deep neural networks for unseen test data. The primary challenge of TTA is limited access to the entire test dataset during online updates, causing error accumulation. To mitigate it, TTA methods have utilized the model output's entropy as a confidence metric that aims to determine which samples have a lower likelihood of causing error. Through experimental studies, however, we observed the unreliability of entropy as a confidence metric for TTA under biased scenarios and theoretically revealed that it stems from the neglect of the influence of latent disentangled factors of data on predictions. Building upon these findings, we introduce a novel TTA method named Destroy Your Object (DeYO), which leverages a newly proposed confidence metric named Pseudo-Label Probability Difference (PLPD). PLPD quantifies the influence of the shape of an object on prediction by measuring the difference between predictions before and after applying an object-destructive transformation. DeYO consists of sample selection and sample weighting, which employ entropy and PLPD concurrently. For robust adaptation, DeYO prioritizes samples that dominantly incorporate shape information when making predictions. Our extensive experiments demonstrate the consistent superiority of DeYO over baseline methods across various scenarios, including biased and wild. Project page is publicly available at https://whitesnowdrop.github.io/DeYO/.

## 1 INTRODUCTION

Although deep neural networks (DNNs) demonstrate powerful performance across various domains, they lack robustness against distribution shifts under conventional training (He et al., 2016; Pan & Yang, 2009). Therefore, research areas such as domain generalization (Blanchard et al., 2011; Gulrajani & Lopez-Paz, 2021), which involves training models to be robust against arbitrary distribution shifts, and unsupervised domain adaptation (UDA) (Ganin & Lempitsky, 2015; Park et al., 2020), which seeks domain-invariant information for label-absent target domains, have been extensively investigated in the existing literature. Test-time adaptation (TTA) (Wang et al., 2021a) has also gained significant attention as a means to address distribution shifts occurring during test time. TTA leverages each data point once for adaptation immediately after inference. Its minimal overhead compared to existing areas makes it particularly suitable for real-world applications (Azimi et al., 2022).

Because UDA assumes access to the entire test samples before adaptation, it utilizes its information on a task by analyzing the distribution of the entire test set (Kang et al., 2019). Oppositely,

---

*Equal Contribution
†Corresponding Authors

due to limited access to the entire test data concurrently, TTA struggles to accurately estimate the entire test data distribution. It leads to inaccurate predictions, and incorporating them into model updates results in error accumulation within the model (Arazo et al., 2020). Hence, it is vital to utilize samples that are less prone to be incorrectly predicted in order to reduce error accumulation. Prior researches (Geifman et al., 2019a; Lee et al., 2022) have adopted the concept of a confidence metric aiming to determine trustworthy samples. Currently, maximum softmax probability (Sohn et al., 2020) and entropy (Saito et al., 2020), utilizing the model's prediction, are the most employed confidence metrics in label-absent tasks. Several TTA methods have also proposed entropy-based sample selection approaches to identify trustworthy samples (Niu et al., 2022; 2023). Although entropy has advantages as a confidence metric, a natural question arises: Can it reliably identify trustworthy samples under various distribution shifts? Notably, we found out that entropy's reliability largely diminishes when a severe spurious correlation shift (Beery et al., 2018) exists in the dataset. To justify the observation, we draw inspiration from Wiles et al. (2022) and elucidate why the sole use of entropy is less reliable for adaptation by introducing latent disentangled factors of inputs.

Based on the observation, we introduce a theoretical proposition for identifying *harmful* samples, those that decrease the model's discriminability during adaptation: a sample is *harmful* if its prediction is more influenced by TRAin-time only Positively correlated with label (TRAP) factors (e.g., background, weather) rather than Commonly Positively-coRrelated with label (CPR) factors (e.g., structure, shape), even if its entropy is low. TRAP factors boost training performance but decrease inference performance, attributable to a discrepancy in the correlation sign with labels. If predictions mainly rely on TRAP factors, there is a high risk of wrong predictions under distribution shifts.

In this paper, we present a new TTA method called Destroy Your Object (DeYO), which leverages our proposed confidence metric, Pseudo-Label Probability Difference (PLPD), to identify *harmful* samples that entropy cannot detect. DeYO uses a single image transformation that distorts the shape of objects, which is a basic element of human visual perception (Geirhos et al., 2019) and is considered a representative CPR factor. PLPD measures the extent to which the probability of pseudo-label decreases after applying the transformation. A high PLPD value indicates that the CPR factor has a large impact on the prediction of the model. DeYO involves two key processes: sample selection and sample weighting. By incorporating PLPD as an additional selection criterion, the model selects samples that are more clearly rooted in CPR factors within the same entropy level for updates. As a result, to emphasize the effect of high-confidence samples on model updates, we assign a greater sample weight to those with low entropy and high PLPD values.

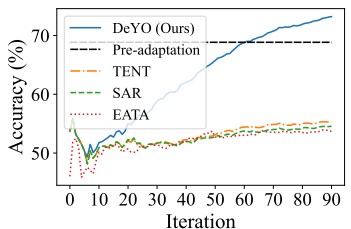

Figure 1: The accuracy within the worst group of the Waterbirds benchmark.

We evaluated DeYO on the ImageNet-C (Hendrycks & Dietterich, 2019) benchmark, not only in the mild scenario but also in wild scenarios, which encompass *mixed shifts*, *label shifts*, and *batch size 1* scenarios. While baseline methods typically exhibit strength in either a mild or wild scenario, DeYO consistently outperforms the baseline methods in all scenarios. Additionally, we tested DeYO on the ColoredMNIST (Arjovsky et al., 2019) and Waterbirds (Sagawa et al., 2020) benchmarks, characterized by an extreme spurious correlations shift, where conventional TTA methods failed to perform adequately (Zhao et al., 2023). In the case of the Waterbirds, as illustrated in Figure 1, baseline methods show lower performance than the pre-adaptation model, whereas DeYO succeeded in showing superiority over it. Furthermore, in the case of the ColoredMNIST, DeYO is the only method that exceeds random guessing (50%) in terms of the worst group accuracy. We conducted diverse analyses to validate the rationale behind DeYO's performance.

**Contributions.** 1) We show that using entropy alone as a measure of confidence is insufficient for TTA. Remarkably, we observed that even (extremely) low-entropy samples can potentially diminish the TTA performance. Motivated by it, we establish a new proposition for identifying *harmful* samples based on disentangled factors. 2) We propose a novel TTA method called DeYO, which leverages both our proposed confidence metric called PLPD and entropy. PLPD examines the influence of CPR factors, which is impossible to discern through entropy alone, via the change in model predictions resulting from an object-destructive transformation applied to objects. 3) We demonstrate that DeYO significantly outperforms existing TTA methods through extensive experiments. The improvement is pronounced in more challenging wild scenarios, and specifically, on the ColoredMNIST, DeYO is the first TTA method that exceeds random guessing.

## 2 REVISITING TTA: FROM THE PERSPECTIVE OF DISENTANGLED FACTORS

Sec. 2 shows that the entropy is not enough for TTA through the following subsections: in Sec. 2.1, we present observations that examine the unreliability of entropy and elucidate its inherent characteristics. Inspired by these insights, Sec. 2.2 offers preliminary concepts and essential notations for further analysis. Subsequently, in Sec. 2.3, we provide an in-depth exploration of why relying solely on entropy may not be considered reliable as a confidence metric.

### 2.1 MOTIVATING OBSERVATIONS

Entropy-based sample selection methods can be found in diverse tasks (Saito et al., 2020; Wang et al., 2022). When a model's probabilistic output closely resembles one-hot encoding, it suggests a high likelihood of a sample being drawn from the training data distribution. These samples, characterized by low entropy, are commonly used as trustworthy samples. In the TTA literature, Niu et al. (2022; 2023) provided empirical evidence of the efficacy of entropy-based sample selection on the ImageNet-C benchmark (Hendrycks & Dietterich, 2019).

DNNs easily leverage spurious features as well as semantically meaningful features, resulting in decreased performance when these spurious correlations are prominent (Beery et al., 2018; Geirhos et al., 2020). For example, butterfly images often co-occur with flowers (Singla & Feizi, 2022), leading the model to misclassify the image without a flower. As highlighted by Wiles et al. (2022), a spurious correlation shift is one of the crucial real-world inspired distribution shifts and commonly exists in datasets with varying degrees. Hence, it is necessary to confirm the reliability of entropy during TTA in the presence of a spurious correlation shift.

In order to furnish empirical results related to a spurious correlation shift, we conducted an analysis of entropy on the Waterbirds (Sagawa et al., 2020). This benchmark enables the manipulation of the degree of a spurious correlation shift between class categories and background, and it is classified into four categories depending on the types of classes and backgrounds. More details for the Waterbirds are provided in Appendix E.2. Remarkably, as shown in Fig. 2(a), we observed that within the worst-performing group, samples with entropy values below the first quartile exhibit lower prediction accuracy compared to other intervals. This observation demonstrates that the application of entropy-based sample selection to adaptation may, instead, result in a performance decline. Indeed, when we applied entropy-based adaptation on the Waterbirds benchmark, as shown in Fig. 1, it resulted in lower performance compared to the pre-adaptation model.

To visually investigate the differences between correct and wrong samples with extremely high confidence, we employ Grad-CAM (Selvaraju et al., 2017) in Fig. 2(b), (c). It reveals that correct samples primarily focus on the birds (target object), while wrong samples relatively focus on the background (spurious feature). Theoretically, Zhou et al. (2021) demonstrated that even in situations with only input distribution shifts (i.e., covariate shifts), deep models also learn spurious features present in the training data. Therefore, relying solely on entropy may not be consistently reliable under distribution shifts, as it cannot distinguish whether the model focuses on the spurious feature.

### 2.2 PRELIMINARIES

In TTA, we have a model $\mathcal{M}_{\boldsymbol{\theta}}$ trained on $\mathcal{D}^{\text{train}} = \{(\mathbf{x}_i^{\text{train}}, y_i^{\text{train}})\}_{i=1}^{N^{\text{train}}}$ with parameter $\boldsymbol{\theta} = \{\theta_i\}_{i=1}^{|\boldsymbol{\theta}|}$, where $\mathbf{x}_i^{\text{train}} \in \mathcal{X}^{\text{train}}$ and $y_i^{\text{train}} \in \mathcal{Y}$. The purpose of TTA is successfully adapting $\mathcal{M}_{\boldsymbol{\theta}}$ using the test data $\mathcal{D}^{\text{test}} = \{(\mathbf{x}_i^{\text{test}}, y_i^{\text{test}})\}_{i=1}^{N^{\text{test}}}$, where $\mathbf{x}_i^{\text{test}} \in \mathcal{X}^{\text{test}}$ and $y_i^{\text{test}} \in \mathcal{Y}$. During adaptation, we cannot access $\mathbf{y}^{\text{test}}$. Instead, existing TTA methods update $\boldsymbol{\theta}$ in the direction of minimizing $\text{Ent}_{\boldsymbol{\theta}}(\mathbf{x}^{\text{test}})$, the entropy of $\mathbf{x}^{\text{test}}$.

$$\text{Ent}_{\boldsymbol{\theta}}(\mathbf{x}) = -\mathbf{p}_{\boldsymbol{\theta}}(\mathbf{x}) \cdot \log \mathbf{p}_{\boldsymbol{\theta}}(\mathbf{x}) = -\sum_{i=1}^{C} \mathbf{p}_{\boldsymbol{\theta}}(\mathbf{x})_i \log \mathbf{p}_{\boldsymbol{\theta}}(\mathbf{x})_i, \tag{1}$$

where $\mathbf{p}_{\boldsymbol{\theta}}(\mathbf{x}) = \text{softmax}(\mathcal{M}_{\boldsymbol{\theta}}(\mathbf{x})) = (\mathbf{p}_{\boldsymbol{\theta}}(\mathbf{x})_1, \dots, \mathbf{p}_{\boldsymbol{\theta}}(\mathbf{x})_C) \in \mathbb{R}^C$ is the model's output probability on $\mathbf{x}$ and $C$ is the number of classes.

Motivated by Wiles et al. (2022), we assume that there is a disentangled latent vector $\mathbf{v}(\mathbf{x}) = (\mathbf{v}_1(\mathbf{x}), \cdots, \mathbf{v}_{d_v}(\mathbf{x})) \in \mathcal{V}$ corresponding to an input $\mathbf{x}$, where $\mathbf{v}_i(\mathbf{x})$ is called $i$-th factor of $\mathbf{x}$.

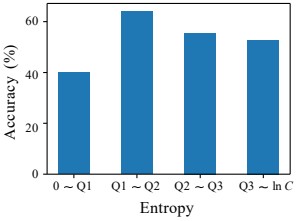 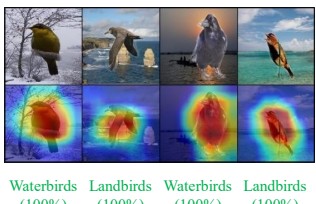 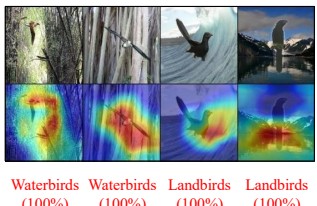

| (a) Entropy level vs accuracy | (b) Grad-CAM of correct samples | (c) Grad-CAM of wrong samples |

Figure 2: (a) A graph that represents accuracy by entropy levels. The lowest entropy interval $0 \sim Q1$ exhibits the lowest accuracy. (b) and (c) display Grad-CAM visualization of samples with correct and incorrect predictions with extremely low entropy, respectively.

For convenience, we will interchangeably use $\mathbf{v}$ and $\mathbf{v}(\mathbf{x})$, assume $v_i \in [0,1]$, and focus on binary classification, where $y \in \{-1, 1\}$. We first define two values: $\text{corr}_i^{\text{train}}$ and $\text{corr}_i^{\text{test}}$, where $\text{corr}_i^{\text{train}} = \text{corr}(y^{\text{train}}, v_i^{\text{train}})$ is the correlation between the train label $y^{\text{train}}$ and the $i$-th factor $v_i^{\text{train}}$ corresponding to $\mathbf{x}^{\text{train}}$, and $\text{corr}_i^{\text{test}} = \text{corr}(y^{\text{test}}, v_i^{\text{test}})$ is the counterpart to $\mathbf{x}^{\text{test}}$. Then we can divide $\mathbf{v}$ into four partitions based on $\text{corr}_i^{\text{train}}$ and $\text{corr}_i^{\text{test}}$:

$$\mathbf{v}_{pp} = \{v_i | \text{corr}_i^{\text{train}} > 0, \text{corr}_i^{\text{test}} > 0\}, \quad \mathbf{v}_{pn} = \{v_i | \text{corr}_i^{\text{train}} > 0, \text{corr}_i^{\text{test}} \le 0\},$$
$$\mathbf{v}_{np} = \{v_i | \text{corr}_i^{\text{train}} \le 0, \text{corr}_i^{\text{test}} > 0\}, \quad \mathbf{v}_{nn} = \{v_i | \text{corr}_i^{\text{train}} \le 0, \text{corr}_i^{\text{test}} \le 0\}. \tag{2}$$

### 2.3 ENTROPY IS NOT ENOUGH

As mentioned in Sec. 2.1, entropy cannot be considered a reliable confidence score in situations involving a spurious correlation shift. In this subsection, we validate the inadequacy of the sole use of entropy as the confidence score from the perspective of disentangled factors. Let us assume that $\mathcal{M}_{\boldsymbol{\theta}}$ is a linear classifier. Then, the parameter $\boldsymbol{\theta}$ also have four partitions $\{\boldsymbol{\theta}_{pp}, \boldsymbol{\theta}_{pn}, \boldsymbol{\theta}_{np}, \boldsymbol{\theta}_{nn}\}$ corresponding to $\{\mathbf{v}_{pp}, \mathbf{v}_{pn}, \mathbf{v}_{np}, \mathbf{v}_{nn}\}$. Then, we can express the logit $a_{\boldsymbol{\theta}}$, probabilistic scalar output $p_{\boldsymbol{\theta}}$, and pseudo-label $\hat{y}$ as follows:

$$a_{\boldsymbol{\theta}}(\mathbf{x}) = \mathcal{M}_{\boldsymbol{\theta}}(\mathbf{x}) = \boldsymbol{\theta} \cdot \mathbf{v}(\mathbf{x}) = \boldsymbol{\theta}_{pp} \cdot \mathbf{v}_{pp} + \boldsymbol{\theta}_{pn} \cdot \mathbf{v}_{pn} + \boldsymbol{\theta}_{np} \cdot \mathbf{v}_{np} + \boldsymbol{\theta}_{nn} \cdot \mathbf{v}_{nn}, \tag{3}$$

$$p_{\boldsymbol{\theta}}(\mathbf{x}) = \sigma(a_{\boldsymbol{\theta}}(\mathbf{x})) = \frac{1}{1 + \exp(-a_{\boldsymbol{\theta}}(\mathbf{x}))}, \quad \hat{y} = \begin{cases} 1 & a_{\boldsymbol{\theta}}(\mathbf{x}) > 0 \\ -1 & \text{otherwise} \end{cases}, \tag{4}$$

where $\sigma(\cdot)$ is the sigmoid function. Then, in the case of TTA, the following proposition holds.

**Proposition 1.** *Let us consider a pre-trained linear classifier $\mathcal{M}_{\boldsymbol{\theta}}$ that uses the latent disentangled factors $\mathbf{v}(\mathbf{x})$ of sample $\mathbf{x}$ as input. We define a **harmful** sample as one that reduces the difference in the mean logits between classes when used for adaptation. A sample $\mathbf{x} \in \mathcal{X}^{\text{test}}$ is a **harmful** sample for adaptation using entropy minimization loss if it satisfies the following condition:*

$$\hat{y} \mathbf{v}(\mathbf{x}) \cdot (\mathbb{E}_{\mathbf{x}^{\text{test}} \sim \mathcal{X}_{+1}^{\text{test}}}[\mathbf{v}(\mathbf{x}^{\text{test}})] - \mathbb{E}_{\mathbf{x}^{\text{test}} \sim \mathcal{X}_{-1}^{\text{test}}}[\mathbf{v}(\mathbf{x}^{\text{test}})]) < 0, \tag{5}$$

*where $\mathcal{X}_y^{\text{test}} = \{\mathbf{x} | (\mathbf{x}, y) \in \mathcal{D}^{\text{test}}, y = y\}$, and $y \in \{1, -1\}$.*

A detailed proof is provided in Appendix A. In the rest of this section, with Proposition 1, we will explain why the samples with low entropy can be harmful.

According to the definition of the partition of disentangled factors, the partitions of optimal parameters $\boldsymbol{\theta}^*$ for the training data satisfy $\boldsymbol{\theta}_{pp}^*, \boldsymbol{\theta}_{pn}^* > 0$, $\boldsymbol{\theta}_{np}^*, \boldsymbol{\theta}_{nn}^* \le 0$. Since $\theta_i$ shares the same sign as $\theta_i^*$ in the early stages of adaptation, $\mathbf{x}$ with a high-confidence pseudo-label of $\hat{y} = 1$ satisfies

$$a_{\boldsymbol{\theta}}(\mathbf{x}) = \boldsymbol{\theta}_{pp} \cdot \mathbf{v}_{pp} + \boldsymbol{\theta}_{pn} \cdot \mathbf{v}_{pn} + \boldsymbol{\theta}_{np} \cdot \mathbf{v}_{np} + \boldsymbol{\theta}_{nn} \cdot \mathbf{v}_{nn} \gg 0,$$
$$|\boldsymbol{\theta}_{pp} \cdot \mathbf{v}_{pp} + \boldsymbol{\theta}_{pn} \cdot \mathbf{v}_{pn}| \gg |\boldsymbol{\theta}_{np} \cdot \mathbf{v}_{np} + \boldsymbol{\theta}_{nn} \cdot \mathbf{v}_{nn}|.$$

In other words, the elements of $\mathbf{v}_{np}$ and $\mathbf{v}_{nn}$ tend to become zero, while $\mathbf{v}_{pp}$ and $\mathbf{v}_{pn}$ become the dominant factors that compose $\mathbf{x}$. We denote $\mathbf{v}_{pp}$ as Commonly Positively-coRrelated with label (CPR) factors and $\mathbf{v}_{pn}$ as TRAin-time only Positively-correlated with label (TRAP) factors. The expected value of $\mathbf{v}(\mathbf{x}^{\text{test}})$ follows the relationship by the defined CPR and TRAP factors:

$$\mathbb{E}_{\mathbf{x}^{\text{test}} \sim \mathcal{X}_{+1}^{\text{test}}}[\mathbf{v}_{pp}(\mathbf{x}^{\text{test}})] > \mathbb{E}_{\mathbf{x}^{\text{test}} \sim \mathcal{X}_{-1}^{\text{test}}}[\mathbf{v}_{pp}(\mathbf{x}^{\text{test}})],$$
$$\mathbb{E}_{\mathbf{x}^{\text{test}} \sim \mathcal{X}_{+1}^{\text{test}}}[\mathbf{v}_{pn}(\mathbf{x}^{\text{test}})] \le \mathbb{E}_{\mathbf{x}^{\text{test}} \sim \mathcal{X}_{-1}^{\text{test}}}[\mathbf{v}_{pn}(\mathbf{x}^{\text{test}})]. \tag{6}$$

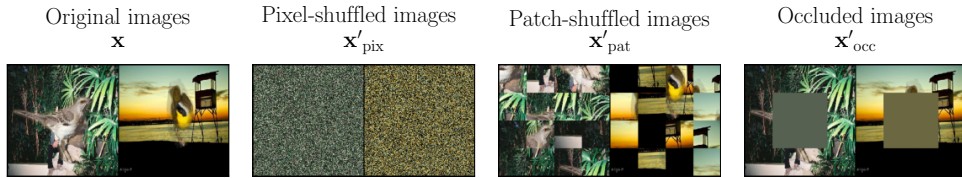

Figure 3: Examples of a transformed image $\mathbf{x}'$ created by different object-destructive transformation methods. $\mathbf{x}$ is an example of WaterBirds.

For $\mathbf{x}$ with a highly confident $\hat{y} = 1$, Eq. 5 can be approximated as follows:

$$\hat{\mathbf{y}}\mathbf{v}(\mathbf{x})\cdot(\mathbb{E}_{\mathbf{x}^{\text{test}}\sim\mathcal{X}_{+1}^{\text{test}}}[\mathbf{v}(\mathbf{x}^{\text{test}})] - \mathbb{E}_{\mathbf{x}^{\text{test}}\sim\mathcal{X}_{-1}^{\text{test}}}[\mathbf{v}(\mathbf{x}^{\text{test}})])$$

$$\approx \underbrace{\mathbf{v}_{pp}(\mathbf{x})\cdot(\mathbb{E}_{\mathbf{x}^{\text{test}}\sim\mathcal{X}_{+1}^{\text{test}}}[\mathbf{v}_{pp}(\mathbf{x}^{\text{test}})] - \mathbb{E}_{\mathbf{x}^{\text{test}}\sim\mathcal{X}_{-1}^{\text{test}}}[\mathbf{v}_{pp}(\mathbf{x}^{\text{test}})])}_{(7.a)} \quad (7)$$

$$+ \underbrace{\mathbf{v}_{pn}(\mathbf{x})\cdot(\mathbb{E}_{\mathbf{x}^{\text{test}}\sim\mathcal{X}_{+1}^{\text{test}}}[\mathbf{v}_{pn}(\mathbf{x}^{\text{test}})] - \mathbb{E}_{\mathbf{x}^{\text{test}}\sim\mathcal{X}_{-1}^{\text{test}}}[\mathbf{v}_{pn}(\mathbf{x}^{\text{test}})])}_{(7.b)} < 0$$

In Eq. 7, as per Eq. 6, (7.a) related to CPR factors becomes positive, while (7.b) related to TRAP factors becomes negative. Therefore, $\mathbf{x}^{C \ll T}$, indicating $\mathbf{x}$ with $\|(7.a)\| \ll \|(7.b)\|$ is a *harmful* sample even if it shows high confidence in terms of entropy. This highlights that entropy, which relies solely on the compressed information expressed as $\boldsymbol{\theta}\cdot\mathbf{v}$, cannot distinguish *harmful* samples by its value. If an adaptation is performed that does not take into account CPR and TRAP factors, in extreme cases, the relative order between two class logits may even change, leading to entirely incorrect predictions. Hence, we aimed to introduce a novel confidence metric that addresses various distribution shifts in the test dataset by avoiding TRAP factors and incorporating CPR factors.

## 3 METHODOLOGY

In Sec. 2, we highlighted the issue of using entropy that ignores the influence of disentangled factors. In Sec. 3, we propose a novel TTA method named Destroy Your Object (DeYO) that incorporates the newly proposed Pseudo-Label Probability Difference (PLPD) score to account for the influence of CPR factors on the model's predictions, particularly the shape information of objects. By integrating the PLPD score that enforces the consideration of CPR factors while suppressing TRAP factors, we alleviate the limitations tied to the exclusive reliance on entropy. Our DeYO consists of sample selection (Sec. 3.1) and weighting (Sec. 3.2) based on the PLPD score.

### 3.1 SAMPLE SELECTION

For sample selection in Niu et al. (2022; 2023), they employ a well-known confidence metric: entropy (Eq. 1). However, as elaborated on in Sec. 2, where there exists a substantial distribution disparity between $\mathcal{X}^{\text{train}}$ and $\mathcal{X}^{\text{test}}$ such as a spurious correlations shift, the entropy becomes highly dependent on the TRAP factors, undermining its reliability. To be robust against distribution shifts, it is crucial to capture CPR factors. While capturing all CPR factors is challenging, we leverage the prominent and certain CPR factor: the shape information of objects (Geirhos et al., 2019).

Incorporating the shape information of objects can be achieved through image transformations such as pixel, patch-shuffling, or center occlusion. As illustrated in Fig. 3, each object destruction technique possesses distinct characteristics. With pixel-shuffling, the mean color of the image is maintained, but it becomes difficult to discern both the object and the background. Patch-shuffling disrupts the shape of the object but preserves local information through the patches. Center occlusion allows for the preservation of the background. However, when the object is not centered or exceptionally large, it may not fully disrupt the object's shape. We conducted experiments with all three techniques, and as illustrated in Sec. 4.3, we observed the best performance with patch-shuffling which solely eliminates the shape information of objects, as opposed to affecting other components.

We propose a novel sample selection strategy based on entropy and PLPD to identify reliable samples for model updates. Our method employs the following sample selection criteria:

$$S_{\boldsymbol{\theta}}(\mathbf{x}) = \{\mathbf{x}|\text{Ent}_{\boldsymbol{\theta}}(\mathbf{x}) < \tau_{\text{Ent}},\ \text{PLPD}_{\boldsymbol{\theta}}(\mathbf{x}, \mathbf{x}') > \tau_{\text{PLPD}}\}, \quad (8)$$

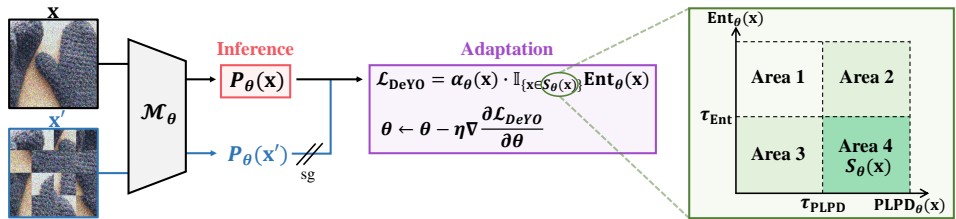

Figure 4: The overview of DeYO. DeYO comprises sample selection (Sec. 3.1) and sample weighting (Sec. 3.2) mechanisms. The areas within the green box are distinguished based on entropy and PLPD intervals, with Area 4 corresponding to $S_{\boldsymbol{\theta}}(\mathbf{x})$ in Sec. 3.1.

$$\mathrm{PLPD}_{\boldsymbol{\theta}}(\mathbf{x}, \mathbf{x}') = (\mathbf{p}_{\boldsymbol{\theta}}(\mathbf{x}) - \mathbf{p}_{\boldsymbol{\theta}}(\mathbf{x}'))_{\hat{y}}, \qquad (9)$$

where $\mathbf{x}$ is an input and $\mathbf{x}'$ is a patch-shuffled input. $\tau_{\mathrm{Ent}}$ and $\tau_{\mathrm{PLPD}}$ are pre-defined thresholds for entropy and PLPD, respectively, and $\hat{y} = \arg\max \mathbf{p}_{\boldsymbol{\theta}}(\mathbf{x})$ is a pseudo-label estimated by the prediction of $\mathbf{x}$. PLPD can be expressed as $(\sigma(\boldsymbol{\theta}\cdot\mathbf{v}) - \sigma(\boldsymbol{\theta}\cdot\mathbf{v} - \boldsymbol{\theta}_{pp}\cdot\mathbf{v}_{pp}))_{\hat{y}}$ under the assumptions in Sec. 2. It represents that entropy-based sample selection does not consider the influence of CPR factors on predictions because it relies on a single prediction. In contrast, PLPD assesses the influence of CPR factors by quantifying a prediction difference contingent on the presence or absence of the shape information. Therefore, the use of PLPD supplements the entropy-based sample selection that may hold samples primarily influenced by TRAP factors. This selection process necessitates only a single additional forward pass, incurring negligible overhead, as it does not require additional back-propagation.

## 3.2 SAMPLE WEIGHTING

Our approach enables identified samples to make varying contributions to the model updates through sample weighting. Specifically, samples belonging to Areas 1, 2, and 3 in Fig. 4 are excluded, yet the contribution of samples in Area 4 to the model update varies based on their reliability. Formally, we express the weighting function $\alpha_{\boldsymbol{\theta}}(\mathbf{x})$ as follows:

$$\alpha_{\boldsymbol{\theta}}(\mathbf{x}) = \frac{1}{\exp\{(\mathrm{Ent}_{\boldsymbol{\theta}}(\mathbf{x}) - \mathrm{Ent}_0)\}} + \frac{1}{\exp\{-\mathrm{PLPD}_{\boldsymbol{\theta}}(\mathbf{x}, \mathbf{x}')\}}, \qquad (10)$$

where $\mathrm{Ent}_0$ is a normalization factor. The first entropy-based weighting term has been employed in existing methods (Wang et al., 2021b; Niu et al., 2022) and is generally effective on varying benchmarks. However, based on the observed unreliability, we introduce an additional PLPD-based weighting term. This term assigns a larger weight as the predictions are more grounded in the object's shape. We confirm that each weighting term is effective, however combining them leads to improved performance, as presented in Sec. 4.3.

## 3.3 OVERALL PROCEDURE OF DEYO

DeYO performs sample selection by exploiting only the samples belonging to $S_{\boldsymbol{\theta}}(\mathbf{x})$ (Eq. 8) and calculates the sample-wise weights $\alpha_{\boldsymbol{\theta}}(\mathbf{x})$ (Eq. 10) to prioritize samples that particularly roots its prediction in the CPR factors. Then, the overall sample-weighted loss is given by:

$$\mathcal{L}_{\mathrm{DeYO}}(\mathbf{x}; \boldsymbol{\theta}) = \alpha_{\boldsymbol{\theta}}(\mathbf{x}) \cdot \mathbb{I}_{\{\mathbf{x} \in S_{\boldsymbol{\theta}}(\mathbf{x})\}} \mathrm{Ent}_{\boldsymbol{\theta}}(\mathbf{x}), \qquad (11)$$

which combines entropy-based and PLPD-based terms in both selection and weighting. The algorithmic representation of DeYO can be found in Algorithm 1 of Appendix B.

## 4 EXPERIMENTS

We designed our experiments to answer the following questions: 1) How does DeYO perform in comparison to baseline methods across various scenarios, including biased and wild scenarios that resemble real-world situations? 2) What role does the newly proposed confidence metric, PLPD, play within DeYO? 3) How do the different components contribute to its performance, and to what degree is DeYO influenced by hyperparameters?

Table 1: Comparisons with baselines on ImageNet-C at severity level 5 under a mild scenario regarding accuracy (%). The **bold** value signifies the top-performing result.

| Mild | Noise | | | Blur | | | | Weather | | | | Digital | | | | Avg. |
|---|---|---|---|---|---|---|---|---|---|---|---|---|---|---|---|---|
| | Gauss. | Shot | Impul. | Defoc. | Glass | Motion | Zoom | Snow | Frost | Fog | Brit. | Contr. | Elastic | Pixel | JPEG | |
| ResNet-50-BN | 2.2 | 2.9 | 1.8 | 17.9 | 9.8 | 14.8 | 22.5 | 16.9 | 23.3 | 24.4 | 58.9 | 5.4 | 16.9 | 20.7 | 31.7 | 18.0 |
| • MEMO | 7.5 | 8.8 | 8.9 | 19.8 | 13.0 | 20.7 | 27.7 | 25.3 | 28.7 | 32.2 | 61.0 | 11.0 | 23.8 | 33.0 | 37.6 | 23.9 |
| • Tent | 29.2 | 31.2 | 30.1 | 28.1 | 27.7 | 41.4 | 49.4 | 47.2 | 41.5 | 57.7 | 67.4 | 29.2 | 54.8 | 58.5 | 52.4 | 43.1 |
| • EATA | 34.9 | 37.1 | 35.8 | 33.4 | 33.0 | 47.1 | 52.7 | 51.6 | 45.7 | 60.0 | **68.1** | 44.4 | 57.9 | 60.6 | 55.1 | 47.8 |
| • SAR | 30.6 | 30.6 | 31.3 | 28.5 | 28.5 | 41.9 | 49.4 | 47.1 | 42.2 | 57.5 | 67.3 | 37.8 | 54.6 | 58.4 | 52.1 | 43.9 |
| • DeYO (ours) | $35.6_{\pm0.2}$ | $37.9_{\pm0.1}$ | $37.1_{\pm0.1}$ | $33.8_{\pm0.2}$ | $34.1_{\pm0.2}$ | $48.5_{\pm0.1}$ | $52.8_{\pm0.1}$ | $52.7_{\pm0.0}$ | $46.4_{\pm0.1}$ | $60.6_{\pm0.0}$ | $68.0_{\pm0.1}$ | $46.1_{\pm0.1}$ | $58.4_{\pm0.1}$ | $61.5_{\pm0.1}$ | $55.7_{\pm0.1}$ | $48.6_{\pm0.0}$ |

Table 2: Comparisons with baselines on ColoredMNIST regarding accuracy (%).

| Biased | Avg Acc | Worst-Group Acc |
|---|---|---|
| ResNet-18-BN | 63.40 | 20.05 |
| • Tent | 57.06 | 9.80 |
| • MEMO | 63.77 | 6.23 |
| • SENTRY | 63.23 | 15.78 |
| • EATA | 60.81 | 17.98 |
| • SAR | 58.37 | 12.36 |
| • DeYO (ours) | **78.24** | **67.39** |

Table 3: Comparisons with baselines on Water-Birds regarding accuracy (%).

| Biased | Avg Acc | Worst-Group Acc |
|---|---|---|
| ResNet-50-BN | 83.16 | 64.90 |
| • Tent | 82.95 | 54.14 |
| • MEMO | 82.34 | 50.47 |
| • SENTRY | 85.77 | 60.90 |
| • EATA | 82.38 | 52.38 |
| • SAR | 82.60 | 53.41 |
| • DeYO (ours) | **87.42** | **73.92** |

**Benchmarks, Test Scenarios, and Models.** For comparison, we conducted experiments on five benchmarks: 1) ImageNet-C (Hendrycks & Dietterich, 2019), a well-known TTA benchmark categorized into 15 corruption types encompassing 5 severity levels for each type, 2) ColoredMNIST and WaterBirds, two benchmarks assessing performance under an extreme spurious correlation shift presented in the dataset, and 3) ImageNet-R (Hendrycks et al., 2021) and VisDA-2021 (Bashkirova et al., 2022), two benchmarks encompassing diverse distribution shifts due to data collected from different style domains (e.g., cartoon, sketch, etc.) compared to ImageNet-C to assess the efficacy for more challenging wild test scenarios. For test scenarios, we followed the mild scenario proposed by Wang et al. (2021a) and three wild test scenarios suggested by Niu et al. (2023). Furthermore, we propose new biased scenarios with ColoredMNIST and WaterBirds, which encompass severe distribution shifts. Regarding the choice of models, we conducted experiments using ResNet-18-BN and ResNet-50-BN (batch normalization), ResNet-50-GN (group normalization), and VitBase-LN (layer normalization), taking into consideration TTA's utilization across various normalization layers. More implementation and baseline details and hyperparameters can be found in Appendix E.

## 4.1 MAIN RESULTS

**Comparison on Mild Scenario.** For the mild scenario, the comparison results on ImageNet-C are reported in Tab. 1. DeYO consistently outperforms the baseline methods across all 15 corruption types in terms of accuracy, affirming the effectiveness of our approach. Notably, even when compared to EATA (Niu et al., 2022), which demonstrates outstanding performance in the mild scenario, DeYO exhibited a 0.8% higher performance. We also confirmed that DeYO yields comparable computational efficiency to EATA and SAR (Niu et al., 2023) as summarized in Appendix C.

**Comparison on Biased Scenario.** Furthermore, DeYO showcased clearly remarkable performance on test benchmarks that are susceptible to being influenced by TRAP factors. As shown in Tab. 2 and 3, we observed substantial performance improvements of 17.43% and 4.47% respectively on ColoredMNIST and WaterBirds, which inherently include a spurious correlation shift, compared to the second best results. For the worst group accuracy on ColoredMNIST, only DeYO surpasses random guessing.

Table 4: Comparisons with baselines on ImageNet-C at severity level 3 and 5 under a mixture of 15 corruption regarding accuracy (%).

| Mixed Shifts | Level 5 | Level 3 |
|---|---|---|
| ResNet-50-GN | 30.6 | 54.0 |
| • MEMO | 31.2 | 54.5 |
| • Tent | 34.2 | 33.1 |
| • EATA | 38.2 | 56.1 |
| • SAR | 38.3 | 57.4 |
| • DeYO (ours) | $38.6_{\pm1.3}$ | $59.2_{\pm0.05}$ |
| VitBase-LN | 29.9 | 53.8 |
| • MEMO | 39.1 | 62.1 |
| • Tent | 24.1 | 70.2 |
| • EATA | 56.4 | 69.6 |
| • SAR | 57.1 | 70.7 |
| • DeYO (ours) | $59.4_{\pm0.1}$ | $72.1_{\pm0.01}$ |

**Comparison on Wild Scenario.** TTA has been demonstrated to enhance the model's robustness against domain shifts. However, its outstanding performance is often achieved under certain mild test conditions. For instance, adapting with a batch of test samples featuring the same type of distribution shift. In complex real-world scenarios, test data can arrive in a more unpredictable manner. Thus, SAR has proposed three more realistic test scenarios: i) dynamic shifts in the ground-truth test label distribution, which leads to imbalanced distributions at each corruption, ii) a single test sample, and iii) a combination of multiple distribution shifts.

Table 5: Comparisons with baselines on ImageNet-C at severity level 5 under online imbalanced label shifts (imbalance ratio = ∞) or under batch size 1 regarding accuracy (%).

| | Noise | | | Blur | | | | Weather | | | | Digital | | | | |
|---|---|---|---|---|---|---|---|---|---|---|---|---|---|---|---|---|
| Label Shifts | Gauss. | Shot | Impul. | Defoc. | Glass | Motion | Zoom | Snow | Frost | Fog | Brit. | Contr. | Elastic | Pixel | JPEG | Avg. |
| ResNet-50-GN | 17.9 | 19.9 | 17.9 | 19.7 | 11.3 | 21.3 | 24.9 | 40.4 | 47.4 | 33.6 | 69.3 | 36.3 | 18.7 | 28.4 | 52.2 | 30.6 |
| • MEMO | 18.4 | 20.6 | 18.4 | 17.1 | 12.7 | 21.8 | 26.9 | 40.7 | 46.9 | 34.8 | 69.6 | 36.4 | 19.2 | 32.2 | 53.4 | 31.3 |
| • Tent | 3.6 | 4.2 | 4.4 | 16.5 | 5.9 | 26.9 | 28.4 | 17.9 | 26.2 | 2.3 | 72.2 | 46.1 | 7.3 | 52.3 | 56.2 | 24.7 |
| • EATA | 25.7 | 28.6 | 24.8 | 18.5 | 19.6 | 24.1 | 28.4 | 35.3 | 33.0 | 41.2 | 65.2 | 33.3 | 28.0 | 42.4 | 43.1 | 32.7 |
| • SAR | 33.7 | 36.9 | 35.3 | 19.3 | 20.3 | 33.8 | 29.8 | 21.9 | 44.7 | 34.9 | 71.9 | 46.7 | 6.6 | 52.3 | 56.2 | 36.3 |
| • DeYO (ours) | $42.5_{\pm0.5}$ | $44.9_{\pm0.2}$ | $43.8_{\pm0.3}$ | $22.2_{\pm0.0}$ | $16.3_{\pm10.2}$ | $41.0_{\pm0.2}$ | $13.2_{\pm9.8}$ | $52.2_{\pm0.4}$ | $51.5_{\pm0.5}$ | $39.7_{\pm27.4}$ | $73.4_{\pm0.1}$ | $52.6_{\pm0.2}$ | $46.9_{\pm1.2}$ | $59.3_{\pm0.1}$ | $59.3_{\pm0.0}$ | $43.9_{\pm2.0}$ |
| VitBase-LN | 9.4 | 6.7 | 8.3 | 29.1 | 23.4 | 34.0 | 27.1 | 15.8 | 26.4 | 47.4 | 54.7 | 44.0 | 30.5 | 44.5 | 47.6 | 29.9 |
| • MEMO | 21.6 | 17.4 | 20.6 | 37.1 | 29.6 | 40.6 | 34.4 | 25.0 | 34.8 | 55.2 | 65.0 | 54.9 | 37.4 | 55.5 | 57.7 | 39.1 |
| • Tent | 33.9 | 1.8 | 27.2 | 54.8 | 52.9 | 58.6 | 54.3 | 12.4 | 11.7 | 69.7 | 76.3 | 66.3 | 59.6 | 69.7 | 66.6 | 47.7 |
| • EATA | 36.2 | 34.7 | 35.5 | 43.4 | 44.3 | 49.3 | 48.5 | 53.2 | 53.5 | 62.3 | 72.7 | 18.8 | 58.0 | 64.7 | 62.8 | 49.2 |
| • SAR | 42.3 | 34.9 | 44.1 | 50.0 | 50.5 | 55.6 | 53.1 | 59.7 | 47.2 | 66.2 | 75.2 | 50.3 | 60.1 | 67.3 | 65.0 | 54.8 |
| • DeYO (ours) | $53.5_{\pm0.5}$ | $36.0_{\pm25.2}$ | $54.6_{\pm0.8}$ | $57.6_{\pm0.2}$ | $58.7_{\pm0.2}$ | $63.7_{\pm0.1}$ | $46.2_{\pm18.7}$ | $67.6_{\pm0.1}$ | $66.0_{\pm0.1}$ | $73.2_{\pm0.2}$ | $77.9_{\pm0.1}$ | $66.7_{\pm0.1}$ | $69.0_{\pm0.1}$ | $73.5_{\pm0.1}$ | $70.3_{\pm0.2}$ | $62.3_{\pm1.7}$ |

| | Noise | | | Blur | | | | Weather | | | | Digital | | | | |
|---|---|---|---|---|---|---|---|---|---|---|---|---|---|---|---|---|
| Batch Size 1 | Gauss. | Shot | Impul. | Defoc. | Glass | Motion | Zoom | Snow | Frost | Fog | Brit. | Contr. | Elastic | Pixel | JPEG | Avg. |
| ResNet-50-GN | 18.0 | 19.8 | 17.9 | 19.8 | 11.4 | 21.4 | 24.9 | 40.4 | 47.3 | 33.6 | 69.3 | 36.3 | 18.6 | 28.4 | 52.3 | 30.6 |
| • MEMO | 18.5 | 20.5 | 18.4 | 17.1 | 12.6 | 21.8 | 26.9 | 40.4 | 47.0 | 34.4 | 69.5 | 36.5 | 19.2 | 32.1 | 53.3 | 31.2 |
| • Tent | 3.1 | 4.2 | 4.0 | 16.5 | 5.3 | 27.4 | 30.3 | 17.7 | 24.9 | 2.0 | 72.1 | 46.2 | 7.8 | 52.6 | 56.3 | 24.7 |
| • EATA | 24.8 | 27.9 | 25.8 | 17.9 | 17.3 | 28.7 | 29.3 | 44.7 | 44.4 | 40.2 | 71.0 | 44.5 | 27.0 | 46.8 | 55.6 | 36.4 |
| • SAR | 23.3 | 26.6 | 23.9 | 18.5 | 15.2 | 28.6 | 30.3 | 44.0 | 44.7 | 29.0 | 72.3 | 44.6 | 13.1 | 46.8 | 56.1 | 34.5 |
| • DeYO (ours) | $41.8_{\pm0.7}$ | $44.7_{\pm0.4}$ | $43.0_{\pm0.7}$ | $22.5_{\pm0.1}$ | $24.7_{\pm0.3}$ | $41.8_{\pm0.1}$ | $24.4_{\pm9.8}$ | $54.5_{\pm0.2}$ | $52.2_{\pm0.1}$ | $20.7_{\pm26.8}$ | $73.5_{\pm0.0}$ | $53.5_{\pm0.2}$ | $48.5_{\pm0.3}$ | $60.2_{\pm0.0}$ | $59.8_{\pm0.1}$ | $44.4_{\pm1.2}$ |
| VitBase-LN | 9.5 | 6.8 | 8.2 | 29.0 | 23.5 | 33.9 | 27.1 | 15.9 | 26.5 | 47.2 | 54.7 | 44.1 | 30.5 | 44.5 | 47.8 | 29.9 |
| • MEMO | 21.6 | 17.3 | 20.6 | 37.1 | 29.6 | 40.4 | 34.4 | 24.9 | 34.7 | 55.1 | 64.8 | 54.9 | 37.4 | 55.4 | 57.6 | 39.1 |
| • Tent | 43.0 | 1.6 | 43.9 | 52.8 | 48.8 | 55.9 | 51.3 | 22.9 | 21.1 | 66.9 | 75.1 | 65.0 | 54.0 | 67.0 | 64.3 | 48.9 |
| • EATA | 32.2 | 26.7 | 30.3 | 43.8 | 40.1 | 47.7 | 42.6 | 35.7 | 43.4 | 60.8 | 65.6 | 61.1 | 46.5 | 60.5 | 58.2 | 46.3 |
| • SAR | 40.6 | 36.9 | 41.9 | 53.7 | 50.5 | 57.4 | 52.8 | 58.9 | 52.7 | 68.9 | 76.0 | 65.8 | 57.9 | 68.9 | 65.8 | 56.6 |
| • DeYO (ours) | $54.0_{\pm0.7}$ | $52.1_{\pm3.6}$ | $55.1_{\pm0.8}$ | $58.8_{\pm0.1}$ | $59.5_{\pm0.1}$ | $64.2_{\pm0.1}$ | $53.5_{\pm5.5}$ | $68.2_{\pm0.1}$ | $66.4_{\pm0.0}$ | $73.7_{\pm0.1}$ | $78.3_{\pm0.0}$ | $68.2_{\pm0.1}$ | $68.9_{\pm0.1}$ | $73.8_{\pm0.1}$ | $70.8_{\pm0.3}$ | $64.4_{\pm0.7}$ |

*i) Online Imbalanced Label Distribution Shifts:* We compared the performance of baseline methods and DeYO in situations where the class imbalance ratio is infinity. As shown in Tab. 5, DeYO exhibited significantly better performance than the baseline methods in both ResNet50-GN and VitBase-LN. For VitBase-LN, except for Zoom where Tent (Wang et al., 2021a) showed the best performance, DeYO showcased its superiority across the 14 corruption types.

*ii) Batch Size 1:* As depicted in Tab. 5, DeYO demonstrated superior performance even in scenarios involving a single test sample. For VitBase-LN, DeYO showcased a performance improvement of 7.8% compared to SAR and notably outperforms the baseline methods on all other corruption types.

*iii) Mixed Distribution Shifts:* We evaluated the performance on a mixture of 15 corruption types at severity levels 5 and 3, as shown in Tab. 4. DeYO still exhibited the most superior performance while the performance improvement may not be as pronounced as in the previous two wild test scenarios.

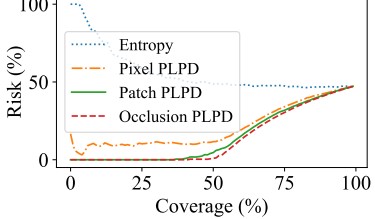

Figure 5: The Risk-Coverage curve of the worst group on Waterbirds.

The significant results are consistently observed on ImageNet-C at severity level 3 (Appendix F) and more challenging benchmarks, ImageNet-R and Visda-2021, belonging to wild test scenarios (Appendix G). The overall results verify that DeYO provides stronger robustness against various distribution shifts. The detailed analysis of PLPD in wild scenarios is provided in Appendix L.

## 4.2 Role and Effect of PLPD

**Discussion on Spurious Correlation.** As evident from Tab. 3, on the Waterbirds benchmark characterized by extreme spurious correlation shifts, DeYO outperformed other baselines. The primary distinction between DeYO and the baselines lies in the inclusion of PLPD as an additional criterion for filtering. To analyze PLPD as a confidence metric, we employ the Risk-Coverage curve (Geifman et al., 2019b), where risk denotes the error rate, and coverage denotes the percentage of input. According to Ding et al. (2020), a reliable confidence metric

| Total 50,000 samples | |
|---|---|
| **Area 1** | **Area 2** |
| 42,095 samples (84.2%) Acc : 9.9% | 1,513 samples (3.0%) Acc : 33.7% |
| **Area 3** | **Area 4** |
| 1,426 samples (2.9%) Acc : 33.5% | 4,966 samples (9.9%) Acc : 49.1% |

Figure 6: Performance and percentage of identified samples on each area on ImageNet-C.

should exhibit a low area under the risk-coverage curve (AURC). As demonstrated in Fig. 5, entropy, in the RC curve of the worst group, yields a large AURC compared to PLPD variations, indicating its limited reliability as a confidence metric. Detailed AURC values can be found in Appendix H. As a result, DeYO significantly outperformed the baselines with entropy filtering.

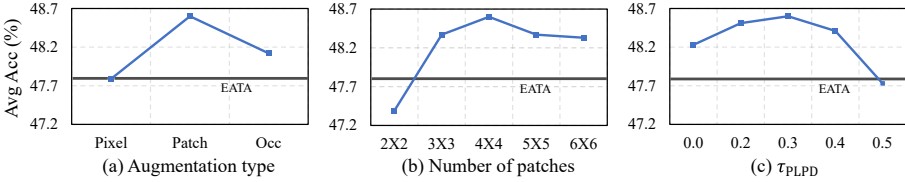

Figure 7: Hyperparameter experiments of DeYO on ImageNet-C at severity level 5.

**Performance of Filtered-out Samples.** While we have established the unreliability of entropy on the Waterbirds benchmark, it is essential to investigate whether PLPD can leverage information that entropy might overlook even on ImageNet-C, which contains a relatively weak spurious correlation shift. This examination is presented in Fig. 6, which illustrates the accuracy of areas in Fig. 4 with the pre-adaptation model. When comparing the accuracy of areas with the same entropy level under Gaussian noise corruption on ImageNet-C at severity level 5, it is evident that Area 2 and Area 4, areas with high PLPD, exhibit higher accuracy compared to Area 1 and Area 3, respectively. This result highlights that PLPD can identify more reliable samples by leveraging information that entropy cannot capture, even in benchmarks with less emphasis on a spurious correlation shift.

## 4.3 HYPERPARAMETER AND ABLATION STUDIES ON DEYO

**Hyperparameter Sensitivity.** We experimented with three transformations: pixel-shuffling, patch-shuffling, and center occlusion, to distort object shapes ($=\mathbf{x}'$). Among these, as shown in Fig. 7(a), patch-shuffling, which eliminates only the shape information while preserving other details, yielded the best performance. When selecting patch-shuffling as transformation, we further tested a different number of patches for an image size of $224 \times 224$. As depicted in Fig. 7(b), the number of patches of $4 \times 4$ exhibited the highest performance, while consistently surpassing EATA from the number of patches of $3 \times 3$ onwards. Lastly, DeYO requires the threshold, $\tau_{\text{PLPD}}$, during the sampling selection process (Fig. 7(c)). A higher $\tau_{\text{PLPD}}$ leads to the removal of more samples, potentially hindering sufficient adaptation knowledge acquisition. We observed that DeYO achieves excellent performance when $\tau_{\text{PLPD}}$ belongs to $[0.2, 0.3]$.

Table 6: Ablation study on the proposed components. Second best = underline.

| | $S_\theta(\mathbf{x})$ | | $\alpha_\theta(\mathbf{x})$ | | |
| --- | --- | --- | --- | --- | --- |
| | $\text{Ent}_\theta$ | $\text{PLPD}_\theta$ | $\text{Ent}_\theta$ | $\text{PLPD}_\theta$ | Avg. |
| (1) Tent | | | | | 43.09 |
| (2) | | | | ✓ | 44.92 |
| (3) | | | ✓ | | 45.92 |
| (4) | | | ✓ | ✓ | 47.06 |
| (5) | | ✓ | | | 44.78 |
| (6) $\text{PLPD}_\theta$ | | ✓ | | ✓ | 46.37 |
| (7) | | ✓ | ✓ | | 47.73 |
| (8) | | ✓ | ✓ | ✓ | 48.44 |
| (9) | ✓ | | | | 44.28 |
| (10) | ✓ | | | ✓ | 46.06 |
| (11) $\text{Ent}_\theta$ | ✓ | | ✓ | | 47.35 |
| (12) | ✓ | | ✓ | ✓ | 47.61 |
| (13) | ✓ | ✓ | | | 44.68 |
| (14) | ✓ | ✓ | | ✓ | 46.38 |
| (15) | ✓ | ✓ | ✓ | | 47.95 |
| (16) DeYO | ✓ | ✓ | ✓ | ✓ | **48.60** |

**Ablation Studies.** To assess the significance of each component of DeYO, we conducted ablation studies on ImageNet-C at severity level 5. Under otherwise identical conditions, using $\text{PLPD}_\theta$ yielded higher performance compared to $\text{Ent}_\theta$, and we achieved the best performance when both $\text{PLPD}_\theta$ and $\text{Ent}_\theta$ are used simultaneously. We also conducted experiments on the biased scenario, WaterBirds. We observed similar trends to ImageNet-C, however, in the case of WaterBirds, we found that utilizing only $\text{PLPD}_\theta$ in $S_\theta$ leads to better performance than utilizing both in $S_\theta$. This validates our observation that entropy significantly decreases in reliability under severe distribution shifts. Further details and analyses of the results are provided in Appendix H.

## 5 CONCLUSION

In contrast to the common consensus, we have theoretically shown that even when utilizing extremely low-entropy samples in adaptation, performance can be degraded if we do not consider the influence of the disentangled factors. Expanding upon the theoretical evidence, we propose DeYO, a TTA method that combines our confidence metric PLPD, designed to account for the influence of the shape information of objects, with entropy. To the best of our knowledge, DeYO is the first TTA method that demonstrates the best performance in both mild and wild scenarios, showing significant performance improvements across various distribution shifts.

ACKNOWLEDGMENTS

This work was supported by the National Research Foundation of Korea (NRF) grants funded by the Korea government (Ministry of Science and ICT, MSIT) (2022R1A3B1077720 and 2022R1A5A708390811), Institute of Information & Communications Technology Planning & Evaluation (IITP) grants funded by the Korea government (MSIT) (2021-0-01343: Artificial Intelligence Graduate School Program (Seoul National University) and 2022-0-00959), and the BK21 FOUR program of the Education and Research Program for Future ICT Pioneers, Seoul National University in 2024.

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

# APPENDIX

## CONTENTS

# A PROOF OF PROPOSITION 1

*Proof.* The gradient when adapting through entropy minimization loss is as follows:

$$
\begin{aligned}
\frac{\partial \text{Ent}_\theta(\mathbf{x})}{\partial \theta_i} &= \frac{\partial \text{Ent}_\theta(\mathbf{x})}{\partial \mathbf{p}_\theta(\mathbf{x})} \cdot \frac{\partial \mathbf{p}_\theta(\mathbf{x})}{\partial \mathbf{a}_\theta(\mathbf{x})} \cdot \frac{\partial \mathbf{a}_\theta(\mathbf{x})}{\partial \theta_i} \\
&= \frac{\partial(-p \log p - (1-p) \log (1-p))}{\partial p} \cdot \frac{\partial \sigma(a)}{\partial a} \cdot \frac{\partial \theta \cdot \mathbf{v}}{\partial \theta_i} \\
&= \log \frac{(1 - \mathbf{p}_\theta(\mathbf{x}))}{\mathbf{p}_\theta(\mathbf{x})} \cdot \sigma(\mathbf{a}_\theta(\mathbf{x}))(1 - \sigma(\mathbf{a}_\theta(\mathbf{x}))) \cdot \mathbf{v}_i \\
&= \text{sign}\left(\log \frac{(1 - \mathbf{p}_\theta(\mathbf{x}))}{\mathbf{p}_\theta(\mathbf{x})}\right) \cdot C \cdot \mathbf{v}_i \\
&= -\hat{\mathbf{y}} \cdot C \cdot \mathbf{v}_i, \text{ where } C = \left|\log \frac{(1 - \mathbf{p}_\theta(\mathbf{x}))}{\mathbf{p}_\theta(\mathbf{x})} \cdot \mathbf{p}_\theta(\mathbf{x})(1 - \mathbf{p}_\theta(\mathbf{x}))\right| > 0,
\end{aligned}
$$

where $p, a$ are substitutes of $\mathbf{p}_\theta(\mathbf{x}), \mathbf{a}_\theta(\mathbf{x})$, respectively. Hence, the update of the model through entropy minimization loss is as follows:

$$
\Delta \theta_i = -\eta \frac{\partial \text{Ent}_\theta(\mathbf{x})}{\partial \theta_i} = \eta \hat{\mathbf{y}} C \mathbf{v}_i = \hat{\mathbf{y}} C' \mathbf{v}_i
$$

$$
\Rightarrow \Delta \theta(\mathbf{x}) = \hat{\mathbf{y}} C' \mathbf{v}(\mathbf{x}), \text{ where } C' = \eta C > 0.
$$

The change in logit of $\mathbf{x}^{\text{test}}$ due to the model's update by $\mathbf{x}$ is as follows:

$$
\Delta(\theta \cdot \mathbf{v}(\mathbf{x}^{\text{test}})) = \Delta \theta(\mathbf{x}) \cdot \mathbf{v}(\mathbf{x}^{\text{test}}) = \hat{\mathbf{y}} C' \mathbf{v}(\mathbf{x}) \cdot \mathbf{v}(\mathbf{x}^{\text{test}}).
$$

The change in the gap between the mean logits of samples belonging to the two classes is as follows:

$$
\begin{aligned}
&\Delta(\mathbb{E}_{\mathbf{x}^{\text{test}} \sim \mathcal{X}^{\text{test}}_{+1}}[\theta \cdot \mathbf{v}(\mathbf{x}^{\text{test}})] - \mathbb{E}_{\mathbf{x}^{\text{test}} \sim \mathcal{X}^{\text{test}}_{-1}}[\theta \cdot \mathbf{v}(\mathbf{x}^{\text{test}})]) \\
&= \mathbb{E}_{\mathbf{x}^{\text{test}} \sim \mathcal{X}^{\text{test}}_{+1}}[\Delta(\theta \cdot \mathbf{v}(\mathbf{x}^{\text{test}}))] - \mathbb{E}_{\mathbf{x}^{\text{test}} \sim \mathcal{X}^{\text{test}}_{-1}}[\Delta(\theta \cdot \mathbf{v}(\mathbf{x}^{\text{test}}))] \\
&= \mathbb{E}_{\mathbf{x}^{\text{test}} \sim \mathcal{X}^{\text{test}}_{+1}}[\Delta \theta \cdot \mathbf{v}(\mathbf{x}^{\text{test}})] - \mathbb{E}_{\mathbf{x}^{\text{test}} \sim \mathcal{X}^{\text{test}}_{-1}}[\Delta \theta \cdot \mathbf{v}(\mathbf{x}^{\text{test}})] \\
&= C' \hat{\mathbf{y}} \mathbf{v}(\mathbf{x}) \cdot (\mathbb{E}_{\mathbf{x}^{\text{test}} \sim \mathcal{X}^{\text{test}}_{+1}}[\mathbf{v}(\mathbf{x}^{\text{test}})] - \mathbb{E}_{\mathbf{x}^{\text{test}} \sim \mathcal{X}^{\text{test}}_{-1}}[\mathbf{v}(\mathbf{x}^{\text{test}})]).
\end{aligned}
$$

If the change is negative, it means that the distribution in the logit space of the two classes is getting closer, leading to a decrease in class discriminability. Since $C'$ is positive, ultimately, when using $\mathbf{x}$ in adaptation that satisfies the following condition, discriminability decreases:

$$
\hat{\mathbf{y}} \mathbf{v}(\mathbf{x}) \cdot (\mathbb{E}_{\mathbf{x}^{\text{test}} \sim \mathcal{X}^{\text{test}}_{+1}}[\mathbf{v}(\mathbf{x}^{\text{test}})] - \mathbb{E}_{\mathbf{x}^{\text{test}} \sim \mathcal{X}^{\text{test}}_{-1}}[\mathbf{v}(\mathbf{x}^{\text{test}})]) < 0.
$$

$\square$

## B PSEUDO CODE OF DEYO

We present the pseudo-code for DeYO, which encompasses several key steps. Firstly, we compute entropy for each test sample $\mathbf{x}_i$ according to Eq. 1. Next, we employ PLPD on the test samples that satisfy the condition specified in lines 4-6 of Algorithm 1. That is, we compute PLPD in conjunction with entropy to select a reliable set denoted as $S_{\boldsymbol{\theta}}(\mathbf{x})$ (Eq. 8). Subsequently, we calculate the weights $\alpha_{\boldsymbol{\theta}}(\mathbf{x})$ for the samples within $S_{\boldsymbol{\theta}}(\mathbf{x})$, taking into account both entropy and PLPD as specified in Eq. 10. Lastly, we optimize the parameter $\tilde{\boldsymbol{\theta}}$ by minimizing $\mathcal{L}_{\text{DeYO}}$, a combination of $\alpha_{\boldsymbol{\theta}}(\mathbf{x})$ and entropy minimization loss as described in Eq. 11. Algorithm 1 provides a detailed pseudo-code representation of these procedures.

---

**Algorithm 1: De**stroy **Y**our **O**bject (DeYO)

**Input:** Test samples $\mathcal{D}^{\text{test}} = \{\mathbf{x}_i\}_{i=1}^{N^{\text{test}}}$, model $\mathcal{M}_{\boldsymbol{\theta}}(\cdot)$ with trainable parameters $\tilde{\boldsymbol{\theta}} \subset \boldsymbol{\theta}$, an object-destructive transformation $\mathcal{A}$, step size $\eta > 0$, hyperparameters $\text{Ent}_0, \tau_{\text{Ent}}, \tau_{\text{PLPD}} > 0$.

**Output:** Predictions $\{\hat{y}_i\}_{i=1}^{N^{\text{test}}}$.

1   Initialize $\tilde{\boldsymbol{\theta}} = \tilde{\boldsymbol{\theta}}_0$;

2   **for** $\mathbf{x}_i \in \mathcal{D}^{\text{test}}$ **do**

3     Compute entropy $\text{Ent}_{\boldsymbol{\theta}}(\mathbf{x}_i)$ and predict $\hat{y}_i = \arg\max_j \mathcal{M}_{\boldsymbol{\theta}}(\mathbf{x}_i)_j$ ;      // (Eq. 1)

4     **if** $Ent_{\boldsymbol{\theta}}(\mathbf{x}_i) > \tau_{Ent}$ **then**

5       |   **continue**;

6     **end**

7     Obtain $\mathbf{x}'_i = \mathcal{A}(\mathbf{x}_i)$ ;

8     Compute $\text{PLPD}_{\boldsymbol{\theta}}(\mathbf{x}_i, \mathbf{x}'_i)$ ;      // (Eq. 9)

9     **if** $PLPD_{\boldsymbol{\theta}}(\mathbf{x}_i, \mathbf{x}'_i) < \tau_{PLPD}$ **then**

10      |   **continue**;

11    **end**

12    Compute sample weight $\alpha_{\boldsymbol{\theta}}(\mathbf{x}_i)$ ;      // (Eq. 10)

13    Compute the overall loss $\mathcal{L}_{\text{DeYO}}$ and its gradient $\nabla_{\tilde{\boldsymbol{\theta}}} \mathcal{L}_{\text{DeYO}}$ ;      // (Eq. 11)

14    Update $\tilde{\boldsymbol{\theta}} \leftarrow \tilde{\boldsymbol{\theta}} - \eta \nabla_{\tilde{\boldsymbol{\theta}}} \mathcal{L}_{\text{DeYO}}$;

15   **end**

---

## C  EFFICIENCY

Table 7: The runtime of state-of-the-art methods. We assess the computational efficiency of various TTA methods using the ResNet-50-BN model on ImageNet-C, specifically examining the case of Gaussian noise at severity level 5, comprising a total of 50,000 images. The practical runtime is evaluated using a single A6000 GPU.

| Method | Need source data? | Online update? | #Forward | #Backward | Other computation | GPU time (50,000 images) |
|---|---|---|---|---|---|---|
| No adapt. | ✗ | ✗ | 50,000 | - | n/a | 72 seconds |
| MEMO | ✗ | ✗ | 50,000×65 | 50,000×64 | AugMix | 30,818 seconds |
| Tent | ✗ | ✓ | 50,000 | 50,000 | n/a | 86 seconds |
| EATA | ✓ | ✓ | 50,000 | 19,085 | regularizer | 97 seconds |
| SAR | ✗ | ✓ | 50,000 + 18,608 | 18,608 + 12,491 | Additional model updates | 123 seconds |
| DeYO (ours) | ✗ | ✓ | 50,000 + 33,943 | 25,836 | Eq. 9 | 114 seconds |

Tab. 7 presents the results of measuring the time required for adaptation in our proposed DeYO and baselines under ImageNet-C, Gaussian noise, and severity level 5 environments. MEMO demands significantly more time due to the necessity of numerous additional augmentations compared to other methods. Generally, other methods show an increase in computation time commensurate with the improvement in performance. DeYO achieves the highest performance while requiring less time than SAR. SAR consumes more GPU time because of its two-step model updates although it has a smaller sum of forward and backward samples than DeYO.

## D  RELATED WORK

### D.1  TEST-TIME ADAPTATION

Test-time adaptation (TTA) (Wang et al., 2021a; Sun et al., 2020; Choi et al., 2022; Lim et al., 2023; Song et al., 2023) aims to enhance inference performance, balancing resource efficiency and effectiveness without access to ground-truth labels. Whereas continual learning (Jung et al., 2023b;a) aims to mitigate distribution shifts occurring while learning a sequence of tasks, TTA aims to efficiently adapt to a target benchmark at inference time by adjusting the pre-trained feature space. Because a ground-truth label is inaccessible in TTA, it requires the use of unsupervised loss to adapt the model. Therefore, entropy minimization serves as a crucial unsupervised regularizer across various tasks (Berthelot et al., 2019; Shu et al., 2018; Liang et al., 2020). By regularizing entropy, we can penalize decisions made in regions of high data density to enhance accuracy for distinct classes.

TTA can be broadly categorized into two groups based on its involvement in the training stage. The first category, known as Test-Time Training (TTT) (Sun et al., 2020), modifies the training loss in the training stage for enhanced test-time adaptation. Typically, TTT involves the simultaneous optimization of the source model using both supervised and self-supervised losses. In other words, TTT relies on a proxy (self-supervised) task, and its loss is contingent upon the selection of a proxy. Depending on the choice of the proxy, TTT may incorporate objectives such as rotation prediction (Gidaris et al., 2018) or contrastive-based losses (Liu et al., 2021b; Bartler et al., 2022).

The second category, Fully Test-Time Adaptation (fully TTA) (Wang et al., 2021a), constitutes a research domain that refrains from intervening in the training stage and, thus, is applicable to all pre-trained models. Our proposed DeYO falls within the realm of fully TTA as it can be applied to any pre-trained model. While fully TTA exhibits distinct advantages compared to TTT, there are limitations in terms of computational cost and stability. Tent (Wang et al., 2021a) utilizes an entropy minimization loss, focusing on updating batch normalization parameters to maintain efficiency while improving performance. Unlike Tent using all samples for adaptation, EATA (Niu et al., 2022) and SAR (Niu et al., 2023) achieved improved performance by filtering-out high entropy samples. However, EATA necessitates a relatively unrealistic assumption of class balance and a clean dataset, and SAR involves a longer computational time due to the additional backward passes. On the other hand, MEMO (Zhang et al., 2022) aims for stable single-sample adaptation by minimizing marginal entropy through multiple augmentations, but it suffers from a longer computational time due to the use of multiple augmentations. DeYO, requiring no additional assumptions and avoiding extra backward passes, achieves significantly improved performance with minimal overhead by utilizing a single augmentation.

## D.2 DISENTANGLED FACTORS

The disentanglement literature (Higgins et al., 2017; Kim & Mnih, 2018; Chen et al., 2018; Dittadi et al., 2021; Lee et al., 2021; 2023) primarily centers around the task of decomposing images into disentangled factors, which represent distinct and independent attributes. This literature operates under the assumption that the data encompasses multiple (potentially numerous) factors and expects models to become invariant to these factors after being exposed to various values of them. This, in turn, enables the models to generalize to novel instances and diverse factor distributions. For instance, in a dataset containing factors like shape and color, the aim is for the model to predict shapes even when confronted with previously unseen colors or different color distributions. Wiles et al. (2022) discusses distribution shifts based on the joint distribution of these disentangled factors. To enhance robustness against such shifts, various methods like weighted resampling (Liu et al., 2021a), data augmentation (Hendrycks et al., 2020), and representation learning (Higgins et al., 2017) are available. Our specific focus lies on weighted resampling, and, to bolster its reliability, we incorporate a single data augmentation step.

## D.3 SELF-TRAINING

Self-training methods have demonstrated state-of-the-art performance in semi-supervised learning (Xie et al., 2020), adversarial robustness (Long et al., 2013), and unsupervised domain adaptation (Shu et al., 2018). Pseudo-labeling and conditional entropy minimization emerge as two prominent forms of self-training. This typically involves supervised training on confidently predicted target pseudo-labels (Tan et al., 2020), or conditional entropy minimization on target instances (Grandvalet & Bengio, 2004). However, unconstrained self-training may result in the accumulation of errors. Consequently, in the field of semi-supervised learning, the error accumulation issues are addressed through using a supervised loss as guidance. In unsupervised domain adaptation, entropy is also employed as a metric to identify reliable target instances (Prabhu et al., 2021).

Despite the success of self-training methods, there exists a limited understanding of the conditions and factors contributing to their effectiveness amid domain shifts. In the theoretical work by Chen et al. (2020b), it is demonstrated that in the particular setting, self-training can avoid using spurious features. While this work contributes to an understanding of self-training in domain shifts, the particular setting discussed, "the spurious feature which correlates with the label in the source domain does not exist in the target domain" and "the source model must be accurately trained", are difficult to be considered as practical.

## D.4 CONSISTENCY UNDER AUGMENTATIONS

The utility of predictive consistency under augmentations has been recognized in various applications, which can be broadly categorized into two main approaches. Firstly, it is employed as a regularizer (Berthelot et al., 2019; Sajjadi et al., 2016). In this context, consistency regularization is applied by introducing data augmentation, capitalizing on the principle that a classifier should yield identical class distributions for an unlabeled example, even post-augmentation. This approach finds extensive applicability in supervised learning (Cubuk et al., 2020), self-supervised representation learning (Chen et al., 2020a), semi-supervised learning (Berthelot et al., 2019; Sajjadi et al., 2016), and unsupervised domain adaptation (Li et al., 2020).

The second approach is centered around the utilization of predictive consistency under augmentations for detecting reliable instances (Prabhu et al., 2021). This approach is rooted in the understanding that consistency under image transformations serves as a dependable indicator of model errors (Bahat et al., 2019). Specifically, in the context of self-training, this approach is employed to mitigate error accumulation. It leverages predictive consistency under a committee of random transforms to identify instances deemed reliable for alignment. Subsequently, the model is selectively optimized on these identified instances.

# E    MORE IMPLEMENTATION DETAILS

## E.1    BASELINE METHODS

We compare DeYO with the following state-of-the-art TTA methods: MEMO (Zhang et al., 2022) enhances the consistency of predictions across various augmented copies for adaptation. SEN-TRY (Prabhu et al., 2021) is an UDA method which enhances target instance reliability by optimizing selective entropy based on consistency under varied image transformations and balances target class distributions using pseudo-labels. For use in the TTA, we remove the supervised loss of SENTRY. Tent (Wang et al., 2021a) guides model updates by reducing the entropy of test samples. EATA (Niu et al., 2022) introduced a novel approach that combines sample selection based on entropy and weighted adjustments to minimize entropy for selected samples. SAR (Niu et al., 2023) minimizes entropy with sharpness awareness for stable adaptation under wild test scenarios. Except for MEMO, the TTA methods including DeYO make real-time (online) adjustments to the model during testing. Once adaptation for each dataset is complete, the model parameters are reset.

## E.2    MORE DETAILS ON BENCHMARK

We assess DeYO's performance across five benchmarks: ImageNet-C (Hendrycks & Dietterich, 2019), ImageNet-R (Hendrycks et al., 2021), Visda-2021 (Bashkirova et al., 2022), ColoredM-NIST (Arjovsky et al., 2019), and Waterbirds (Sagawa et al., 2020). ImageNet-C, created by applying 15 distinct corruptions to the ImageNet (Krizhevsky et al., 2012) benchmark, is widely employed to conduct experiments observing robustness against shifts in distribution. For the evaluation of DeYO under more challenging scenarios, we extend our analysis beyond the ImageNet-C benchmark by including ImageNet-R (Hendrycks et al., 2021) and Visda-2021 (Bashkirova et al., 2022) benchmarks. ImageNet-R consists of a diverse array of artistic renditions (e.g., art, cartoons, graffiti, sculptures, etc.) representing 200 ImageNet classes. The Visda-2021 benchmark collects data from both ImageNet-R/C and ObjectNet (Barbu et al., 2019). Unlike ImageNet-C, which primarily focuses on synthetic corruption, ImageNet-R, and Visda-2021 present more challenges due to the natural shifts inherent in the data domain.

For the ColoredMNIST and Waterbirds benchmarks, the training data exhibits substantial spurious correlation, whereas the test data is characterized by minimal or no spurious correlation. The ColoredMNIST benchmark is a derived version of the MNIST (LeCun et al., 1998). In this benchmark, images with digits ranging from 0 to 4 are assigned to class 0, while those with digits from 5 to 9 are assigned to class 1. The color ID is green when it is 0 and red when it is 1. The default color ID follows the class ID and is flipped with a probability of 0.2 on the training data and 0.9 on the test data. The Waterbirds benchmark combines objects from the CUB (Wah et al., 2011) with backgrounds from the Places (Zhou et al., 2017). The correlations between the class and background are present in the training data but not in the test data. In the training set, 95% of waterbird images are positioned on water backgrounds, while the remaining 5% are placed on land backgrounds. Similarly, 95% of landbird images are on land backgrounds, with 5% positioned on water backgrounds.

## E.3    MORE DETAILS ON EXPERIMENTAL PROTOCOLS

For the ColoredMNIST benchmark, we opt for ResNet-18-BN, given its lower complexity compared to other benchmarks. For Waterbirds and ImageNet-C (mild scenario), we employ ResNet-50-BN, while for ImageNet-C (wild scenarios), ImageNet-R, and Visda-2021, we employ ResNet-50-GN and ViTbase-LN. Pre-training for ColoredMNIST and Waterbirds involves 20 and 200 epochs, respectively, with a batch size of 64. As for ImageNet-C, ImageNet-R, and Visda-2021, we utilize publicly available pre-trained models from the torchvision and timm (Wightman, 2019) libraries. Our testing adaptation protocol basically adheres to the one proposed by Niu et al. (2023). We optimize only normalization layer parameters. Under label shift scenarios of ImageNet-R and Visda-2021, we assume an imbalance ratio of infinity, randomizing the order of classes and samples within each class. Stochastic Gradient Descent (SGD) with a momentum of 0.9 serves as our optimizer. Typically, the batch size is set to 64, except for experiments where it is configured as 1 (batch size 1). The learning rate is fixed at 0.00025 for the ResNet model and 0.001 for the Vit model, except when the batch size equals 1. In such cases, the learning rate is adjusted to 0.00025 divided by 16

for the ResNet model and 0.001 divided by 32 for the Vit model. For the Waterbirds benchmark, DeYO filters out a substantial number of samples, so we increase the learning rate by a factor of 5.

The required hyperparameters for DeYO are $\tau_{Ent}$, $\tau_{PLPD}$, and $Ent_0$. We set $Ent_0$ and $\tau_{Ent}$ to $0.4 \times \ln C$ and $0.5 \times \ln C$, respectively. The analysis of two hyperparameters ($\tau_{Ent}$, and $Ent_0$) is presented in Appendix H, and the analysis of $\tau_{PLPD}$ sensitivity can be found in Section 4.3 of the main paper. As shown in the sensitivity results, DeYO exhibits robustness with respect to hyperparameters. Regarding $\tau_{PLPD}$ for the reported results, its value is configured at 0.3 for ImageNet-C (mild scenario), 0.5 for the biased scenario, and 0.2 for the rest of the experiments. ColoredMNIST and Waterbirds are binary benchmarks, and since they mostly produce one-hot predictions due to their simplicity, we use 0.5 for a threshold of label flip. We do not use entropy filtering and set $Ent_0 = \ln C$ for ColoredMNIST.

## F   FURTHER EXPERIMENTS ON IMAGENET-C AT SEVERITY LEVEL 3

The results at severity level 3 under the mild scenario and two wild scenarios (online imbalanced label distribution shift, and batch size 1) are reported in Tab. 8 and 9, respectively. The results are consistent with those of severity level 5, demonstrating superior performance across the 15 corruption types on average.

### F.1   COMPARISONS ON MILD SCENARIO

Table 8: Comparisons with baselines on ImageNet-C at severity level 3 under a mild scenario regarding accuracy (%). The **bold** value signifies the top-performing result.

| | Noise | | | Blur | | | | Weather | | | | Digital | | | | |
|---|---|---|---|---|---|---|---|---|---|---|---|---|---|---|---|---|
| Mild | Gauss. | Shot | Impul. | Defoc. | Glass | Motion | Zoom | Snow | Frost | Fog | Brit. | Contr. | Elastic | Pixel | JPEG | Avg. |
| ResNet-50-BN | 27.6 | 25.0 | 25.2 | 37.9 | 16.9 | 37.7 | 35.2 | 35.2 | 32.1 | 46.7 | 69.6 | 46.0 | 55.6 | 46.2 | 59.3 | 39.7 |
| • Tent | 54.8 | 54.3 | 53.7 | 49.2 | 46.5 | 58.8 | 57.7 | 55.9 | 48.6 | 65.8 | 72.1 | 67.1 | 69.4 | 67.5 | 65.9 | 59.2 |
| • EATA | 57.0 | 56.8 | 56.2 | 52.4 | 50.2 | 61.0 | 59.7 | 58.6 | 51.3 | 67.1 | **72.2** | 68.2 | **69.9** | 68.1 | **66.7** | 61.0 |
| • SAR | 54.6 | 54.1 | 53.5 | 49.3 | 46.3 | 58.6 | 57.6 | 55.6 | 48.6 | 65.6 | 72.0 | 67.1 | 69.3 | 67.3 | 65.7 | 59.0 |
| • DeYO (ours) | **58.1**±0.1 | **58.0**±0.1 | **57.1**±0.1 | **53.4**±0.0 | **51.2**±0.1 | **61.9**±0.1 | **59.8**±0.0 | **59.6**±0.0 | **51.9**±0.2 | **67.6**±0.1 | 72.0±0.1 | **68.5**±0.0 | 69.8±0.1 | **68.6**±0.0 | 66.6±0.0 | **61.6**±0.0 |

### F.2   COMPARISONS ON WILD SCENARIO

Table 9: Comparisons with baselines on ImageNet-C at severity level 3 under online imbalanced label shifts (imbalance ratio = ∞) or under batch size 1 regarding accuracy (%).

| | Noise | | | Blur | | | | Weather | | | | Digital | | | | |
|---|---|---|---|---|---|---|---|---|---|---|---|---|---|---|---|---|
| Label Shifts | Gauss. | Shot | Impul. | Defoc. | Glass | Motion | Zoom | Snow | Frost | Fog | Brit. | Contr. | Elastic | Pixel | JPEG | Avg. |
| ResNet-50-GN | 54.5 | 52.9 | 53.1 | 44.4 | 21.2 | 49.8 | 39.3 | 54.9 | 54.1 | 55.8 | 75.3 | 69.7 | 59.6 | 59.7 | 66.4 | 54.1 |
| • MEMO | 55.9 | 54.3 | 54.1 | 40.1 | 23.1 | 49.5 | 41.4 | 54.8 | 54.1 | 57.6 | 75.7 | 70.2 | 60.2 | 61.5 | 66.7 | 54.6 |
| • Tent | 59.1 | 58.6 | 58.3 | 39.0 | 27.9 | 54.7 | 41.1 | 51.3 | 41.4 | 62.0 | 75.2 | 70.1 | 62.3 | 63.7 | 66.4 | 55.4 |
| • EATA | 52.3 | 52.9 | 51.7 | 35.7 | 30.1 | 46.4 | 39.6 | 43.8 | 39.8 | 55.7 | 72.4 | 66.6 | 54.7 | 56.0 | 56.2 | 50.3 |
| • SAR | 60.8 | 60.5 | 60.2 | 47.9 | 36.7 | 58.2 | 49.7 | 57.9 | 53.6 | 65.0 | 76.4 | 71.0 | 67.0 | 65.8 | 67.6 | 59.9 |
| • DeYO (ours) | **64.0**±0.2 | **63.9**±0.1 | **63.2**±0.2 | **54.0**±0.3 | **44.9**±0.7 | **62.2**±0.1 | **55.1**±0.2 | **61.2**±0.2 | **57.9**±0.4 | **69.2**±0.1 | **76.9**±0.0 | **73.2**±0.1 | **71.2**±0.1 | **70.2**±0.1 | **69.8**±0.0 | **63.8**±0.0 |
| VitBase-LN | 51.5 | 46.8 | 50.4 | 48.7 | 37.1 | 54.7 | 41.6 | 35.1 | 33.3 | 68.0 | 69.3 | 74.9 | 65.9 | 66.0 | 63.6 | 53.8 |
| • MEMO | 62.1 | 57.9 | 61.5 | 57.2 | 45.6 | 62.0 | 49.9 | 46.5 | 43.1 | 74.1 | 75.8 | **79.7** | 72.6 | 72.3 | 70.6 | 62.1 |
| • Tent | 68.7 | 68.0 | 68.1 | 68.2 | 63.8 | 70.9 | 63.8 | 67.6 | 41.9 | 76.3 | 78.8 | 79.5 | 75.9 | 76.7 | 73.7 | 69.5 |
| • EATA | 65.3 | 62.6 | 63.6 | 63.0 | 57.1 | 66.3 | 59.3 | 64.5 | 61.0 | 73.3 | 76.9 | 75.9 | 74.2 | 74.8 | 73.1 | 67.4 |
| • SAR | 68.8 | 68.2 | 68.4 | 68.3 | 64.7 | 71.0 | 64.2 | 68.1 | 66.0 | 76.4 | 79.0 | 79.6 | 76.2 | 77.1 | 74.1 | 71.3 |
| • DeYO (ours) | **71.7**±0.1 | **71.6**±0.1 | **71.4**±0.1 | **70.6**±0.1 | **68.9**±0.1 | **73.9**±0.1 | **69.2**±0.2 | **72.4**±0.2 | **69.7**±0.1 | **77.9**±0.2 | **80.2**±0.1 | 79.5±0.0 | **78.2**±0.1 | **78.7**±0.0 | **76.7**±0.2 | **74.0**±0.1 |

| | Noise | | | Blur | | | | Weather | | | | Digital | | | | |
|---|---|---|---|---|---|---|---|---|---|---|---|---|---|---|---|---|
| Batch Size 1 | Gauss. | Shot | Impul. | Defoc. | Glass | Motion | Zoom | Snow | Frost | Fog | Brit. | Contr. | Elastic | Pixel | JPEG | Avg. |
| ResNet-50-GN | 54.5 | 52.8 | 53.1 | 44.3 | 21.2 | 49.7 | 39.2 | 54.8 | 54.0 | 55.8 | 75.4 | 69.8 | 59.6 | 59.7 | 66.3 | 54.0 |
| • MEMO | 55.7 | 54.2 | 53.9 | 40.0 | 22.8 | 49.2 | 41.2 | 54.8 | 54.1 | 57.6 | 75.5 | 69.9 | 60.0 | 61.3 | 66.6 | 54.5 |
| • Tent | 58.8 | 58.5 | 58.7 | 38.2 | 26.8 | 54.9 | 42.6 | 51.6 | 38.8 | 61.9 | 75.3 | 70.0 | 62.3 | 63.6 | 66.3 | 55.2 |
| • EATA | 59.2 | 58.7 | 58.8 | 45.7 | 32.6 | 55.5 | 45.9 | 56.4 | 52.7 | 63.6 | 75.9 | 71.1 | 64.7 | 64.5 | 67.8 | 58.2 |
| • SAR | 60.3 | 59.6 | 59.5 | 46.6 | 33.0 | 57.5 | 47.8 | 57.8 | 52.8 | 65.1 | 76.7 | 71.4 | 67.3 | 66.0 | 67.8 | 59.3 |
| • DeYO (ours) | **64.4**±0.1 | **64.5**±0.1 | **63.7**±0.1 | **55.2**±0.1 | **46.0**±0.1 | **63.1**±0.1 | **55.9**±0.3 | **62.3**±0.2 | **58.8**±0.2 | **69.8**±0.1 | **77.0**±0.1 | **73.5**±0.2 | **71.5**±0.0 | **70.7**±0.1 | **70.2**±0.1 | **64.5**±0.0 |
| VitBase-LN | 51.6 | 46.9 | 50.5 | 48.7 | 37.2 | 54.7 | 41.6 | 35.1 | 33.5 | 67.8 | 69.3 | 74.8 | 65.8 | 66.0 | 63.7 | 53.8 |
| • MEMO | 61.9 | 57.7 | 61.4 | 57.0 | 45.4 | 61.8 | 49.8 | 46.6 | 43.1 | 73.9 | 75.7 | 79.6 | 72.6 | 72.1 | 70.5 | 61.9 |
| • Tent | 67.1 | 66.2 | 66.3 | 66.3 | 60.9 | 69.1 | 61.4 | 65.2 | 60.4 | 75.2 | 78.1 | 78.8 | 74.9 | 75.8 | 72.4 | 69.2 |
| • EATA | 60.7 | 58.5 | 61.6 | 60.1 | 51.8 | 64.2 | 54.8 | 53.3 | 52.6 | 72.5 | 73.6 | 77.9 | 71.3 | 71.3 | 69.7 | 63.6 |
| • SAR | 68.5 | 67.8 | 68.0 | 67.8 | 63.1 | 70.7 | 63.5 | 66.9 | 62.8 | 75.8 | 77.7 | 78.4 | 74.7 | 75.7 | 72.7 | 70.3 |
| • DeYO (ours) | **72.3**±0.1 | **72.1**±0.1 | **71.9**±0.0 | **71.1**±0.0 | **69.4**±0.0 | **74.2**±0.0 | **69.3**±0.1 | **72.8**±0.1 | **70.1**±0.1 | **78.6**±0.0 | **80.7**±0.0 | **80.4**±0.0 | **78.6**±0.0 | **79.2**±0.0 | **77.2**±0.1 | **74.5**±0.0 |

## G   FURTHER EXPERIMENTS ON IMAGENET-R AND VISDA-2021

We conducted additional experiments for wild scenarios using the ImageNet-R and Visda-2021 benchmarks with ResNet-50-GN and ViTBase-LN. Both benchmarks involve collecting data from various distribution shifts, including data with completely different styles aside from corruption. Therefore, for overall experiments, we consistently assumed mixed shift scenarios and then included label shifts or batch size 1 scenarios.

Tab. 10 and 11 report the results on online imbalanced label distribution shifts and batch size 1 scenarios, respectively. Similar to the results in the main for ImageNet-C, DeYO exhibited the best performance across scenarios and architectures on the ImageNet-R benchmark.

Table 10: Comparisons with baselines on ImageNet-R under a wild scenario (online imbalanced label distribution shifts) regarding accuracy (%).

| Method | ResNet-50-GN | VitBase-LN |
|---|---|---|
| No Adapt. | 40.8 | 43.1 |
| Tent | 42.8 | 49.9 |
| EATA | 42.8 | 51.0 |
| SAR | 42.5 | 51.7 |
| DeYO (ours) | **47.0** | **59.3** |

Table 11: Comparisons with baselines on ImageNet-R under a wild scenario (single sample adaptation (batch size 1)) regarding accuracy (%).

| Method | ResNet-50-GN | VitBase-LN |
|---|---|---|
| No Adapt. | 40.8 | 43.1 |
| Tent | 44.3 | 50.4 |
| EATA | 41.8 | 46.7 |
| SAR | 43.1 | 52.0 |
| DeYO (ours) | **48.4** | **60.3** |

We also conducted similar experiments on the Visda-2021 benchmark. Tab. 12 and 13 report the results on online imbalanced label distribution shifts and batch size 1 scenarios, respectively. Similarly, DeYO demonstrated the best performance across scenarios and architectures on Visda-2021.

Table 12: Comparisons with baselines on Visda-2021 under a wild scenario (online imbalanced label distribution shifts) regarding accuracy (%).

| Method | ResNet-50-GN | VitBase-LN |
|---|---|---|
| No Adapt. | 43.5 | 44.3 |
| Tent | 43.8 | 50.2 |
| EATA | 43.2 | 51.1 |
| SAR | 43.7 | 50.4 |
| DeYO (ours) | **44.9** | **57.3** |

Table 13: Comparisons with baselines on Visda-2021 under a wild scenario (single sample adaptation (batch size 1)) regarding accuracy (%).

| Method | ResNet-50-GN | VitBase-LN |
|---|---|---|
| No Adapt. | 43.5 | 44.3 |
| Tent | 43.9 | 50.4 |
| EATA | 43.9 | 47.9 |
| SAR | 43.8 | 51.2 |
| DeYO (ours) | **45.9** | **58.7** |

An important observation in each of these tables is that, despite the inclusion of entropy-based sample selection in EATA and SAR, the performance improvement is minimal or even leads to a decrease in performance. This verifies that the effectiveness of entropy-based sample selection is either negligible or, in some cases, negative for the datasets with various distribution shifts. DeYO demonstrated that significant performance gains are not solely reliant on entropy and that the use of PLPD with consideration for CPR factors contributes to improved adaptation.

## H   FURTHER HYPERPARAMETER AND ABLATE RESULTS

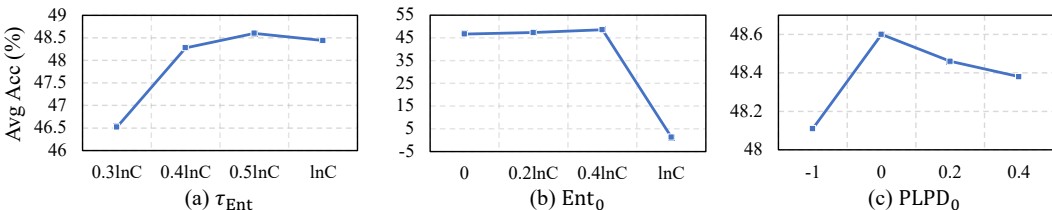

Figure 8: Hyperparameter ($\tau_{Ent}$, $Ent_0$, and $PLPD_0$) experiments on ImageNet-C at severity level 5.

DeYO relies on three essential hyperparameters: $\tau_{\text{Ent}}$, $\tau_{\text{PLPD}}$, and $\text{Ent}_0$. As shown in the main paper, the results for $\tau_{\text{PLPD}}$ indicate that DeYO with $\tau_{\text{PLPD}}$ below 0.4 performs better than EATA, as long as they do not overly restrict sample selection. For $\tau_{\text{Ent}}$, as depicted in Fig. 8(a), there are quite numerous samples not selected by $\tau_{\text{PLPD}}$, making the choice of $0.5 \times \ln C$ more effective than the original $0.4 \times \ln C$ used by EATA. However, notably, even with $0.4 \times \ln C$, DeYO outperformed EATA. Regarding $\text{Ent}_0$ (Fig. 8(b)), it performed best when set to $0.4 \times \ln C$, identical to EATA. Lastly, $\text{PLPD}_0$ represents the normalization factor of the PLPD term in sample weighting. As the best performance was achieved when set to 0, we opted not to include the normalization factor when proposing the term, as shown in Fig. 8(c).

Tab. 14 presents the results of a AURC measurement aimed at assessing the reliability of entropy and PLPD variants as a confidence score in the worst group of the Waterbirds benchmark. Entropy exhibits the highest AURC value, making it the most unreliable confidence metric, while among the PLPD variants, the use of $\mathbf{x}'_{\text{occ}}$ (Occlusion PLPD) demonstrates the lowest value which means the highest reliability. This observation is attributed to the fact that the Waterbirds inherently involve objects precisely centered in the image. Furthermore, an ablation study was conducted to investigate the effects of DeYO's components on Waterbirds, and the results are documented in Tab. 15. While the overall trends align with the findings in Tab. 6, notably, the absence of entropy filtering in case (8) yields higher performance than our proposed DeYO. It additionally illuminates the notion that entropy lacks reliability in the presence of spurious correlation shifts.

Table 14: The AURC value of Fig. 5.

| | AURC (%) |
|---|---|
| Entropy | 56.30 |
| 4×AugMix PLPD | 35.58 |
| Pixel PLPD | 20.71 |
| Patch PLPD | 14.75 |
| Occlusion PLPD | 13.77 |

Table 15: Ablation study on the proposed components on Waterbirds in biased scenario (ResNet-50-BN). Second best = underline.

| | $S_\theta(\mathbf{x})$ $\text{Ent}_\theta$ | $S_\theta(\mathbf{x})$ $\text{PLPD}_\theta$ | $\alpha_\theta(\mathbf{x})$ $\text{Ent}_\theta$ | $\alpha_\theta(\mathbf{x})$ $\text{PLPD}_\theta$ | Avg. |
|---|---|---|---|---|---|
| (1) Tent | | | | | 55.29 |
| (2) | | | | ✓ | 58.46 |
| (3) | | | ✓ | | 55.23 |
| (4) | | | ✓ | ✓ | 59.97 |
| (5) | | ✓ | | | 66.58 |
| (6) $\text{PLPD}_\theta$ | | ✓ | | ✓ | 72.88 |
| (7) | | ✓ | ✓ | | 66.42 |
| (8) | | ✓ | ✓ | ✓ | **75.95** |
| (9) | ✓ | | | | 54.87 |
| (10) | ✓ | | | ✓ | 56.90 |
| (11) $\text{Ent}_\theta$ | ✓ | | ✓ | | 54.87 |
| (12) | ✓ | | ✓ | ✓ | 57.57 |
| (13) | ✓ | ✓ | | | 62.62 |
| (14) | ✓ | ✓ | | ✓ | 70.33 |
| (15) | ✓ | ✓ | ✓ | | 63.30 |
| (16) DeYO | ✓ | ✓ | ✓ | ✓ | 73.14 |

Table 16: Ablation study on the proposed components on ImageNet-C in label shifts scenario (ViTBase-LN). Second best = underline.

| | $S_\theta(\mathbf{x})$ $\text{Ent}_\theta$ | $S_\theta(\mathbf{x})$ $\text{PLPD}_\theta$ | $\alpha_\theta(\mathbf{x})$ $\text{Ent}_\theta$ | $\alpha_\theta(\mathbf{x})$ $\text{PLPD}_\theta$ | Avg. |
|---|---|---|---|---|---|
| (1) Tent | | | | | 52.71 |
| (2) | | | | ✓ | 54.78 |
| (3) | | | ✓ | | 54.04 |
| (4) | | | ✓ | ✓ | 54.08 |
| (5) | | ✓ | | | 59.98 |
| (6) $\text{PLPD}_\theta$ | | ✓ | | ✓ | **61.74** |
| (7) | | ✓ | ✓ | | 61.02 |
| (8) | | ✓ | ✓ | ✓ | 61.44 |
| (9) | ✓ | | | | 52.71 |
| (10) | ✓ | | | ✓ | 56.18 |
| (11) $\text{Ent}_\theta$ | ✓ | | ✓ | | 52.15 |
| (12) | ✓ | | ✓ | ✓ | 54.21 |
| (13) | ✓ | ✓ | | | 59.96 |
| (14) | ✓ | ✓ | | ✓ | 61.58 |
| (15) | ✓ | ✓ | ✓ | | 60.78 |
| (16) DeYO | ✓ | ✓ | ✓ | ✓ | 61.30 |

# I FURTHER HYPERPARAMETER SENSITIVITY RESULTS

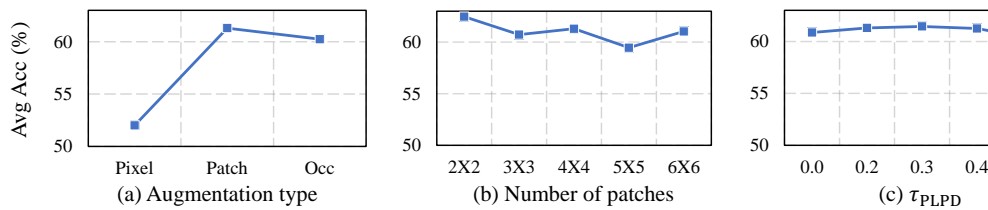

Figure 9: Hyperparameter sensitivity experiments of DeYO on ImageNet-C at severity level 5 under online imbalanced label shifts (imbalance ratio = $\infty$).

As shown in Fig. 9 and Tab. 17, we experimented with the parameter sensitivity of DeYO under a different model architecture (ViTBase), dataset (WaterBirds), and test scenario (wild scenario: online imbalanced label shifts and biased scenario: spurious correlation). We observed consistent results with that of Fig. 7 (ResNet-50-BN, ImageNet-C, and mild scenario) in the main manuscript. For Waterbirds, it involves a 2-class classification task. Therefore, the value of $\tau_{\text{PLPD}}$ should be larger than that of ImageNet-C, which has 1000 classes, to adequately capture the changes induced by transformations.

Table 17: Hyperparameter sensitivity experiments of DeYO on WaterBirds in terms of augmentation type (left), number of patches (center), and $\tau_{\text{PLPD}}$ (right).

| Type | Avg (%) | Worst (%) |
|------|---------|-----------|
| Pixel | 85.84 | 65.42 |
| Patch | 88.16 | 74.18 |
| Occ | 87.69 | 73.55 |

| #Patch | Avg (%) | Worst (%) |
|--------|---------|-----------|
| 2X2 | 86.98 | 74.96 |
| 3X3 | 87.26 | 75.43 |
| 4X4 | 88.16 | 74.18 |
| 5X5 | 87.12 | 72.14 |
| 6X6 | 87.40 | 72.14 |

| $\tau_{\text{PLPD}}$ | Avg (%) | Worst (%) |
|---------|---------|-----------|
| 0.0 | 84.54 | 58.84 |
| 0.2 | 87.31 | 71.67 |
| 0.3 | 87.72 | 72.93 |
| 0.4 | 87.45 | 71.67 |
| 0.5 | 88.16 | 74.18 |
| 0.6 | 88.06 | 74.8 |
| 0.7 | 87.59 | 74.02 |

## J  EFFECT OF MULTIPLE OBJECT-PRESERVING AUGMENTATIONS

We compared the average PLPD measured by augmenting $\mathbf{x}$ multiple times through Aug-Mix (Hendrycks et al., 2020) with the PLPD of pixel-shuffling, patch-shuffling, and center occlusion. AugMix employs various object-preserving augmentations such as autocontrast, equalize, rotate, and solarize, randomly applying them to create multiple augmented instances. Then, the augmented instances are mixed with the original input. AugMix is also utilized in MEMO.

When using the average PLPD for a total of 4 mixed inputs in DeYO, it showed lower performance than all three object-destructive transformation methods in both ColoredMNIST and Waterbirds as shown in the Tab. 18. The AURC, which evaluates the reliability of PLPD, also exhibited worse performance compared to the object-destructive transformations as shown in Tab. 14. Our proposed PLPD needs to disrupt the shape of the object to function effectively. Through the AURC values, we empirically confirmed that AugMix (object-preserving augmentation technique) cannot contribute to measuring the impact on the CPR factor.

We conducted experiments in EATA and DeYO using average entropy over four AugMix augmentations for entropy filtering. As indicated in the Tab. 19, this modification resulted in only marginal performance improvements. The results show that the use of average entropy from augmented images is not beneficial to enhancing robustness against distribution shifts between training and inference phases.

Additionally, in DeYO, we experimented using multiple AugMix($\mathbf{x}$) as $\mathbf{x}'$ for calculating PLPD. AugMix employs various object-preserving augmentations such as autocontrast, equalize, rotate, and solarize, randomly applying them to create multiple augmented instances. Then, the augmented instances are mixed with the original input. AugMix is also utilized in MEMO.

Table 18: The effect of multiple object-preserving augmentations on PLPD.

| | ColoredMNIST | | Waterbirds | |
|---|---|---|---|---|
| | Avg Acc (%) | Worst Acc (%) | Avg Acc (%) | Worst Acc (%) |
| DeYO with $8\times \mathbf{x}'_{\text{AugMix}}$ | 53.53 | 4.19 | 82.94 | 60.75 |
| DeYO with $\mathbf{x}'_{\text{pix}}$ | 69.68 | 50.78 | 85.11 | 64.17 |
| DeYO with $\mathbf{x}'_{\text{occ}}$ | 74.56 | 62.06 | 86.78 | 74.96 |
| DeYO with $\mathbf{x}'_{\text{pat}}$ | 77.61 | 65.51 | 86.56 | 74.18 |

Table 19: The effect of multiple object-preserving augmentations on EATA and DeYO.

| | ColoredMNIST | | Waterbirds | |
|---|---|---|---|---|
| | Avg Acc (%) | Worst Acc (%) | Avg Acc (%) | Worst Acc (%) |
| EATA | 60.29 | 16.99 | 83.42 | 57.48 |
| EATA with $4\times$ AugMix($\mathbf{x}$) | 60.29 | 17.00 | 83.60 | 57.94 |
| DeYO | 77.61 | 65.51 | 86.56 | 74.18 |
| DeYO with $4\times$ AugMix($\mathbf{x}$) | 77.83 | 65.58 | 86.81 | 74.45 |

## K    EFFECT OF PLPD DURING PRE-TRAINING

We experimented with utilizing PLPD-based filtering during pre-training. For Waterbirds, during a total of 200 epochs, the initial 50 epochs were trained conventionally, followed by 150 epochs where only samples with PLPD greater than the threshold of 0.2 were used for pre-training. As shown in Tab. 20, after implementing PLPD filtering, the performance is 5.82% higher compared to the original result on Tent. Similarly, the performance of DeYO also improves by 3.48%.

Table 20: The effect of PLPD during pre-training stage. With a PLPD filtering, the model shows performance improvements after TENT and DeYO.

|  | Tent | | DeYO | |
| --- | --- | --- | --- | --- |
|  | Avg Acc (%) | Worst Acc (%) | Avg Acc (%) | Worst Acc (%) |
| pre-training w/o PLPD | 82.98 | 52.90 | 86.74 | 74.18 |
| pre-training w/ PLPD | 84.45 | 58.72 | 87.90 | 77.66 |

## L    EFFECT OF PLPD IN WILD SCENARIOS

Essentially, the enhancement in performance arising from excluding TRAP factors during adaptation, thereby increasing robustness against covariate shifts and spurious correlations in the wild scenarios (label shifts and batch size 1), is the same as the mild scenario.

Label shifts and batch size 1 scenarios have the following characteristics.

*i) Label shifts:*

- An Input data stream is dependent on the ground truth class.

- When i.i.d. samples are present within a batch (a mild scenario), it's possible to discern which disentangled factors are emphasized for each class. However, in a label shifts scenario, since all samples in a batch belong to the same class, this cannot be determined.

*ii) Batch size 1:*

- A model only uses a single image to update its parameter.

- When i.i.d. samples are present within a batch (a mild scenario), sample-specific noise is reduced through loss averaging. However, in a batch size 1 scenario, this noise cannot be reduced, increasing sample dependency.

According to our theoretical analysis, the increase in the weight of disentangled factors is proportional to the value of the corresponding factors, as described in Appendix A. In general, without PLPD filtering, during updates with batches where all samples have the ground truth label $c$ in a label shifts scenario, the weight of disentangled factors common to samples in the batch increases proportionally to the corresponding factors' values. In a batch size 1 scenario, the weight of all disentangled factors in the current sample increases proportionally to the values of the corresponding factors.

Using only entropy, weights for class $c$ still increase regardless of its CPR and TRAP factors. Consequently, in samples with a different ground truth $c' \neq c$, the increased weights related to class $c$ result in larger logits of $c$, raising the likelihood of an incorrect prediction as class $c$. To maintain the prediction of samples with ground truth $c'$, the value of updated weights' corresponding factors should be small to reduce logit changes. Without prior knowledge of distribution shifts, we infer that CPR factors of class $c$ hold smaller values in class $c'$. Thus, updating only the weights corresponding to CPR factors of class $c$ can preserve the prediction. Utilizing PLPD filtering, which focuses on samples where the influence of CPR factors outweighs that of TRAP factors, allows for the selective update of weights corresponding to CPR factors of class $c$ while minimizing the change in weights for TRAP factors. Therefore, using PLPD filtering can preserve the original predictions of samples with different ground truths.

Compared to a mild scenario, wild scenarios suffer from a lack of diversity in batch information of the distribution of disentangled factors or even proceed with updates using a single image. It exacerbates optimization stability and provokes stochasticity issues. Consequently, the drawbacks of entropy are more pronounced in wild scenarios than in mild ones, leading to a significant performance decline of baseline methods. In contrast, information on CPR/TRAP factors, which only DeYO discriminates, is less affected by the batch size or class dependency of an input stream compared to entropy, offering improved robustness.

## M    LIMITATIONS

Identifying all the CPR factors present in an input image is practically impossible. In other words, achieving a perfect PLPD is practically impossible. Therefore, we focused on one of the fundamental components of human visual perception – the shape of objects – for a reliable PLPD. In cases where a model provides accurate predictions based on specific local features, these predictions remain unaltered even after patch-shuffling augmentation, resulting in a low PLPD value. Therefore, it might be less useful to identify beneficial samples for TTA through a high PLPD strategy. Through the ablation studies, it is evident that PLPD-based sample selection has an advantage over entropy. However, observing that entropy retains its strength in sample weighting, it appears that the effects of PLPD in sample weighting can be explored. These limitations suggest moving beyond the current approach, which relies solely on the last layer (softmax). We think that exploring the intermediate feature maps makes it possible to obtain other CPR factors in different hierarchies.

