# OpenReview forum: "Entropy is not Enough for Test-Time Adaptation: From the Perspective of Disentangled Factors"
_ICLR.cc/2024/Conference — ICLR 2024 spotlight_

### Official Review · Reviewer_NrgV · 2023-10-29

**Soundness:** 2 fair
**Presentation:** 2 fair
**Contribution:** 2 fair
**Rating:** 8
**Confidence:** 5

**Summary:**

TTA approaches select good samples for TTA using entropy as a metric (some of the approaches, and not all as I point in my review). However, a sample can have low entropy but using spurious features for prediction. Including such samples for TTA can be hurtful as they use spurious features for low entropy. The authors propose patch shuffle augmentation to identify such samples and remove them.
The paper has missing important baseline for some core experiments (Table 2 and Table 3), and it is not clear why the approach helps in WILD distribution shift (label shift, batch size 1).

**Strengths:**

Strengths:
- The problem is well motivated i.e. spurious features can cause entropy to be low.

**Weaknesses:**

Weakness:
- Unclear what the authors mean by “disentangled latent vector” exactly for each image. No proper reference has been made as well to get details. It would be better to write more explicitly how does Equation 6 arise for fluency of the reading flow. Authors mention $x^{T>>C}$, please describe this notation (and might be not even required to introduce this as it is clear authors mean second term is more pronounced).
- It is not a great practice to include the whole of related works in the appendix. Even the related works section on TTA in appendix is extremely sparse (a single paragraph!) and should be much more broader in it’s scope.
- I understand that authors aim is to reject samples which use spurious features for prediction as doing TTA on them might be harmful. However, past theoretical work (https://proceedings.neurips.cc/paper/2020/file/f1298750ed09618717f9c10ea8d1d3b0-Paper.pdf) shows that self training avoids using spurious features.
- There are many methods which assess augmentation + entropy (see MEMO, SENTRY https://arxiv.org/pdf/2012.11460.pdf). SENTRY does sample filtering based on consistency of predictions over various augmentations. It is a crucial and missing baseline in this work, especially in Table 2 and Table 3. I know SENTRY operates in UDA setting, but the consistency over augmentations part is pretty standard.
- The authors mention existing works use entropy for selective minimization. There should be a baseline where we do selection based on average entropy over multiple augmentations. It is a trivial extension of previous works and should have been a baseline to asses the effectiveness of pseudo-label confidence drop proposed in this work.
- Why is the MEMO baseline missing in Table 2 and Table 3, where I would guess MEMO to be most effective as it also does entropy average over multiple transformations. It is quite possible that the average entropy over augmentations is low for spurious correlations domainted prediction, making those samples contribute much less to the loss anyways.
- Can the authors give any reason why their proposed approach should work better under label shift or for batch size 1 as shown in Table 5? I do not see any reason why augmentation based sample selection should help for label shift TTA (makes sense for spurious TTA). Similar is the question for Batch size 1 setting. Are the authors sure they did proper hyperparameter tuning for the baselines?
- Further, the effect of such spurious correlations can be removed by incorporating such transforms (cutmix, random cropping, or the used patch shuffling) during pretraining itself. Will the proposed approach help in those cases as well?


Overall, the paper does consider an important question. However, misses some very important baselines. Further, some of the results on some benchmarks have been left unexplained beyond simply stating the numbers. The writing also lacks heavily, with a complete miss on the related works section.


_______post rebuttal_____
Authors addressed some of my major concerns with respect to the baselines. I am increasing my rating from 5 to 6.

**Questions:**

See the weakness section for questions to answer during rebuttal

---

> ### Author Response · Authors · 2023-11-16
> **Response to Reviewer NrgV (1/3)**
>
> Thank you for recognizing the importance of the problem addressed in our paper. Firstly, based on your comments, we were able to improve the readability of the paper. Secondly, leveraging the papers you introduced, we could further demonstrate the robustness and effectiveness of DeYO by incorporating additional theoretical explanations and experiments. Lastly, we agree that there was a lack of intuitive descriptions for the performance of DeYO in wild scenarios. Following your comments, we added in-depth explanations about DeYO. We are truly grateful for your meticulous guidance on our work.
>
> > W1. Unclear what the authors mean by “disentangled latent vector” exactly for each image. No proper reference has been made as well to get details. It would be better to write more explicitly how does Equation 6 arise for fluency of the reading flow. Authors mention $x^{T \gg C}$, please describe this notation?
>
> A: Sorry for the lack of explanation. Following the approach of Olivia et al. [1], we decompose the data into a set of independent random variables. For example, an image of a human face can be decomposed into independent variables representing features like eyes, nose, and mouth. We refer to these, potentially numerous, independent $d_v$ random variables as factors, forming a disentangled latent vector in a $d_v$-dimensional space.
>
> In Eq. (6), by definition, CPR factors are positively correlated with the label. Consequently, the average of CPR factors ($v_{pp}$) in samples with a label of 1 should be greater than that in samples with a label of -1. The same process applies to TRAP factors ($v_{pn}$). We have incorporated the explanation of "$x^{T \gg C}$" into the manuscript as well. Thank you for the valuable feedback.
>
> > W2. It is not a great practice to include the whole of related works in the appendix. Even the related works section on TTA in appendix is extremely sparse (a single paragraph!) and should be much more broader in it’s scope.
>
> A: Given our focus on demonstrating the effectiveness and robustness of DeYO, we overlooked incorporating sufficient content on related work. Following your comment, we revised the related work section to provide a more comprehensive and in-depth overview. Due to space constraints, we have not been able to include this in the main manuscript yet. However, in the final version, we will make an effort to incorporate at least some of it into the main manuscript.
>
> > W3. I understand that authors aim is to reject samples which use spurious features for prediction as doing TTA on them might be harmful. However, past theoretical work (https://proceedings.neurips.cc/paper/2020/file/f1298750ed09618717f9c10ea8d1d3b0-Paper.pdf) shows that self training avoids using spurious features.
>
> A: Thank you for recommending the paper. We included the recommended paper in related work. However, this paper does not seem practically feasible. Self-training requires conditions like “the spurious feature which correlates with the label in the source domain does not exist in the target domain” to avoid using spurious features. Moreover, the source model must be accurately trained. However, it is challenging for the real source model (pre-trained model) and target dataset to satisfy these conditions. Thus, the theoretical proof presented in the recommended paper may not be fully applicable in real-world scenarios.
>
> Also, we empirically confirmed that the performance of existing TTA methods that utilize self-training (entropy minimization) without considering spurious features, is significantly lower than that of DeYO, which deliberately suppresses the learning of spurious features during self-training.

---

> ### Author Response · Authors · 2023-11-16
> **Response to Reviewer NrgV (2/3)**
>
> > W4. There are many methods which assess augmentation + entropy (see MEMO, SENTRY https://arxiv.org/pdf/2012.11460.pdf). SENTRY does sample filtering based on consistency of predictions over various augmentations. It is a crucial and missing baseline in this work, especially in Tab. 2 and 3. I know SENTRY operates in UDA setting, but the consistency over augmentations part is pretty standard.
>
> > W6. Why is the MEMO baseline missing in Tab. 2 and 3, where I would guess MEMO to be most effective as it also does entropy average over multiple transformations. It is quite possible that the average entropy over augmentations is low for spurious correlations domainted prediction, making those samples contribute much less to the loss anyways.
>
> A: Thank you for the constructive comment. We added the results of MEMO and SENTRY to Tabs. 2 and 3 in the main manuscript. We implement SENTRY based on the implementation of MEMO and excluding the supervised loss component following the TTA setting. As shown in the results, SENTRY yields higher performance than MEMO. However, their performance is similar to other baselines and is significantly lower than DeYO.
>
> We think that there are two reasons why SENTRY showed a higher performance than MEMO. Firstly, SENTRY utilizes additional memory to estimate class distribution and employs a loss function based on it. Secondly, instead of identifying confident samples using the average entropy of predictions, SENTRY identifies samples that are not influenced by augmentations.
>
> The reason why the performance of MEMO and SENTRY is significantly lower than DeYO is due to the fundamental issue we highlighted in Sec. 2.1 regarding entropy: “relying solely on entropy may not be consistently reliable in the presence of distribution shifts, as it cannot distinguish whether the model focuses on the desired target.”
> In other words, entropy cannot distinguish whether a sample is primarily represented by CPR factors or TRAP factors, regardless of its value (including those samples contributing much more to the loss you mentioned). This issue cannot be resolved by averaging entropy over multiple augmentations which is used by MEMO and SENTRY because they rely solely on entropy. To address this, we propose a new metric called PLPD, which can measure the influence of CPR factors. We revised the first paragraph of page 5 in the main manuscript to highlight and provide further explainations for the entropy's problem in distinguishing **harmful** samples.
>
> ### MEMO & SENTRY
> ||ColoredMNIST||WaterBirds||
> |-|:-:|:-:|:-:|:-:|
> ||Avg Acc (%)|Worst Acc (%)|Avg Acc (%)|Worst Acc (%)|
> |MEMO|63.77|6.23|82.34|50.47|
> |SENTRY|63.23|15.78|85.77|60.90|
> |TENT|56.28|8.93|82.61|53.43|
> |EATA|60.29|16.99|83.42|57.48|
> |DeYO with $x'_{pat}$|77.61|65.51|86.56|74.18|
>
> > W5. The authors mention existing works use entropy for selective minimization. There should be a baseline where we do selection based on average entropy over multiple augmentations. It is a trivial extension of previous works and should have been a baseline to asses the effectiveness of pseudo-label confidence drop proposed in this work.
>
> A: We agree with your comment that the baseline you proposed can provide a further demonstration of the effectiveness of DeYO. Following your suggestion, we conducted experiments in EATA and DeYO using average entropy over four AugMix [2] augmentations for entropy filtering. As indicated in the table below, this modification resulted in only marginal performance improvements. The results show that the use of average entropy from augmented images is not beneficial to enhancing robustness against distribution shifts between training and inference phases.
>
> ### The effect of multiple augmentations
> ||ColoredMNIST||WaterBirds||
> |-|:-:|:-:|:-:|:-:|
> ||Avg Acc (%)|Worst Acc (%)|Avg Acc (%)|Worst Acc (%)|
> |EATA|60.29|16.99|83.42|57.48|
> |EATA with 4*AugMix($x$)|60.29|17.00|83.6|57.94|
> |DeYO|77.61|65.51|86.56|74.18|
> |DeYO with 4*AugMix($x$)|77.83|65.58|86.81|74.45|
>
> Additionally, in DeYO, we experimented using multiple AugMix($x$) as $x’$ for calculating PLPD. AugMix employs various object-preserving augmentations such as autocontrast, equalize, rotate, and solarize, randomly applying them to create multiple augmented instances. Then, the augmented instances are mixed with the original input. AugMix is also utilized in MEMO.
>
> When using the average PLPD for a total of 8 mixed inputs in DeYO, it showed lower performance than all three object-destructive transformation methods in both ColoredMNIST and Waterbirds. The AURC, which evaluates the reliability of PLPD, also exhibited worse performance compared to the object-destructive transformations. Our proposed PLPD needs to disrupt the shape of the object to function effectively. Through the AURC values, we empirically confirmed that AugMix (object-preserving augmentation technique) cannot contribute to measuring the impact on the CPR factor.

---

> ### Author Response · Authors · 2023-11-16
> **Response to Reviewer NrgV (3/3)**
>
> ### AugMix with PLPD
> ||ColoredMNIST||WaterBirds||
> |-|:-:|:-:|:-:|:-:|
> ||Avg Acc (%)|Worst Acc (%)|Avg Acc (%)|Worst Acc (%)|
> |DeYO with $8*x'_{AugMix}$|53.53|4.19|82.94|60.75|
> |DeYO with $x'_{pix}$|69.68|50.78|85.11|64.17|
> |DeYO with $x'_{occ}$|74.56|62.06|86.78|74.96|
> |DeYO with $x'_{pat}$|77.61|65.51|86.56|74.18|
>
> ### Evaluation of the reliability of AugMix PLPD
> ||Waterbirds|
> |-|:-:|
> ||AURC (%)
> |Entropy|56.3|
> |Pixel PLPD|20.71|
> |Patch PLPD|14.75|
> |Occ PLPD|13.77|
> |4*AugMix PLPD|35.58|
>
> > W7. Can the authors give any reason why their proposed approach should work better under label shift or for batch size 1 as shown in Tab. 5? I do not see any reason why augmentation based sample selection should help for label shift TTA (makes sense for spurious TTA). Similar is the question for Batch size 1 setting. Are the authors sure they did proper hyperparameter tuning for the baselines?
>
> A: Sorry for missing an intuitive description of why our proposed method performs well in wild scenarios compared to existing approaches. The reason why DeYO excels in wild scenarios is that, for the first time, it constrains the direction of adaptation through entropy minimization in distribution shift situations by focusing on the shape of the object. Even in cases where ground-truth labels exist, such as in cross-entropy learning with label shift or batch size 1, training can be disrupted by bias (label shift) or variance (batch size 1). Furthermore, in situations like TTA, error accumulation becomes more severe. Existing methods in such scenarios aim to improve performance by enhancing training stability rather than restricting the direction of adaptation. However, this only delays the worsening of severe error accumulation once it has begun and does not fundamentally reduce it.
>
> We have conducted sufficient hyperparameter tuning for the baselines, and it is already demonstrated at SAR that Tent and EATA excel only in mild scenarios.
>
> > W8. Further, the effect of such spurious correlations can be removed by incorporating such transforms (cutmix, random cropping, or the used patch shuffling) during pretraining itself. Will the proposed approach help in those cases as well?
>
> A: Thank you for suggesting the experiment. This experiment allowed us to further validate the scalability of our proposed method. In line with your comment, we experimented with utilizing PLPD-based filtering during pretraining. For Waterbirds, during a total of 200 epochs, the initial 50 epochs were trained conventionally, followed by 150 epochs where only samples with PLPD greater than the threshold of 0.2 were used for pretraining. After implementing PLPD filtering, the performance is 5.82% higher compared to the original result on Tent. Similarly, the performance of DeYO also improves by 3.48%.
>
> ### Pretraining with PLPD filtering
> |Waterbirds|Tent||DeYO||
> |-|:-:|:-:|:-:|:-:|
> ||Avg Acc (%)|Worst Acc (%)|Avg Acc(%)|Worst Acc(%)|
> |Pretraining w/o PLPD|82.98|52.9|86.64|74.18|
> |Pretraining w/ PLPD|84.45|58.72|87.90|77.66|
>
> > References
> > [1] Wiles, Olivia, et al. "A Fine-Grained Analysis on Distribution Shift." International Conference on Learning Representations. 2022.
> [2] Hendrycks, Dan, et al. "AugMix: A Simple Data Processing Method to Improve Robustness and Uncertainty." International Conference on Learning Representations. 2020.
>
> The final version of the manuscript will incorporate all the responses.

---

> ### Author Response · Authors · 2023-11-21
> **Looking forward to your post-rebuttal feedback!**
>
> Dear Reviewer NrgV,
>
> Thank you again for the insightful comments and suggestions! Given the limited time remaining, we eagerly anticipate your subsequent feedback. It would be our pleasure to offer more responses to further demonstrate the effectiveness of our methodology.
>
> In our previous response, we have thoroughly reviewed your comments and provided responses summarized as follows:
>
> - Addressed writing deficiencies by clarifying unclear terms and expanding the related work section. Our revised related work section covers much broader scopes, including test-time adaptation, disentangled factors, self-training (referencing Self-training Avoids Using Spurious Features Under Domain Shift [1]), and consistency under augmentations (referencing SENTRY [2]) based on your suggestions.
> - Explained why self-training cannot avoid using the spurious features in the real world.
> - Conducted the new experiments (including MEMO and SENTRY) to show the effectiveness of object-destructive transformations compared to common data augmentation methods.
> - Provided more explanations as to why DeYO is still effective in wild scenarios.
> - Conducted the new experiment to show the effectiveness of sample selection based on PLPD during pretraining.
>
> We hope that the provided new experiments and the additional explanation have convinced you of the merits of this paper. If there are additional questions, please feel free to let us know.
>
> Additionally, we wish to express our gratitude once again to you for your insightful feedback. Incorporating your suggestions has undoubtedly enhanced the clarity and robustness of our work.
>
> We deeply appreciate your time and effort!
>
> Best regards, Authors
>
> > References
> [1] Chen, Yining, et al. "Self-training avoids using spurious features under domain shift." Neural Information Processing Systems. 2020.
> [2] Prabhu, Viraj, et al. "Sentry: Selective entropy optimization via committee consistency for unsupervised domain adaptation." International Conference on Computer Vision. 2021.

---

> > ### Comment · Reviewer_NrgV · 2023-11-21
> > **Thanks**
> >
> > I am amazed by the amount of efforts the authors have put in to address my concerns. I will be raising my rating.
> >
> > Can the authors further clarify on why their approach works in label shift and batch size = 1. I didnt understand what they are trying to hint.
> >
> > >The reason why DeYO excels in wild scenarios is that, for the first time, it constrains the direction of adaptation through entropy minimization in distribution shift situations by focusing on the shape of the object. Even in cases where ground-truth labels exist, such as in cross-entropy learning with label shift or batch size 1, training can be disrupted by bias (label shift) or variance (batch size 1).
> >
> > How does focusing on shape relate to helping resolve label shift? I would encourage authors to put in more thoughts on this and give a proper justification. Current justification doesn't make sense or else maybe I am missing something. If I understand why their approach works so well as given my numbers, I can raise my rating further. My only issue now with this paper is I don't understand the source of their gains. In general, I believe this paper should lacks ablation studies.

---

> ### Author Response · Authors · 2023-11-22
> **Response by authors**
>
> Firstly, we are pleased that our responses have provided sufficient answers and have been helpful in enhancing your understanding. We are also pleased to address an additional question, as it contributes to a deeper comprehension of DeYO.
> > Q1: Can the authors further clarify on why their approach works in label shift and batch size 1?
>
> A: To clarify on why our DeYO works well in label shifts and batch size 1, we provide an interpretation more in line with our theoretical analysis. Essentially, the enhancement in performance arising from excluding TRAP factors during adaptation, thereby increasing robustness against covariate shifts and spurious correlations in the wild scenarios (label shifts and batch size 1), is the same as the mild scenario.
>
> Label shifts and batch size 1 scenarios have the following characteristics.
>
> 1. label shifts
>     - An Input data stream is dependent on the ground truth class.
>     - When i.i.d. samples are present within a batch (a mild scenario), it's possible to discern which disentangled factors are emphasized for each class. However, in a label shifts scenario, since all samples in a batch belong to the same class, this cannot be determined.
> 2. batch size 1
>     - A model only uses a single image to update its parameter.
>     - When i.i.d. samples are present within a batch (a mild scenario), sample-specific noise is reduced through loss averaging. However, in a batch size 1 scenario, this noise cannot be reduced, increasing sample dependency.
>
> According to our theoretical analysis, the increase in the weight of disentangled factors is proportional to the value of the corresponding factors, as described in Appendix A. In general, without PLPD filtering, during updates with batches where all samples have the ground truth label $c$ in a label shifts scenario, the weight of disentangled factors common to samples in the batch increases proportionally to the corresponding factors' values. In a batch size 1 scenario, the weight of all disentangled factors in the current sample increases proportionally to the values of the corresponding factors.
>
> Using only entropy, weights for class $c$ still increase regardless of its CPR and TRAP factors. Consequently, in samples with a different ground truth $c’ \ne c$, the increased weights related to class $c$ result in larger logits of $c$, raising the likelihood of an incorrect prediction as class $c$. To maintain the prediction of samples with ground truth $c’$, the value of updated weights' corresponding factors should be small to reduce logit changes. Without prior knowledge of distribution shifts, we infer that CPR factors of class $c$ hold smaller values in class $c’$. Thus, updating only the weights corresponding to CPR factors of class $c$ can preserve the prediction. Utilizing PLPD filtering, which focuses on samples where the influence of CPR factors outweighs that of TRAP factors, allows for the selective update of weights corresponding to CPR factors of class $c$ while minimizing the change in weights for TRAP factors. Therefore, using PLPD filtering can preserve the original predictions of samples with different ground truths.
>
> Compared to a mild scenario, wild scenarios suffer from a lack of diversity in batch information of the distribution of disentangled factors or even proceed with updates using a single image. It exacerbates optimization stability and provokes stochasticity issues. Consequently, the drawbacks of entropy are more pronounced in wild scenarios than in mild ones, leading to a significant performance decline of baseline methods. In contrast, information on CPR/TRAP factors, which only DeYO discriminates, is less affected by the batch size or class dependency of an input stream compared to entropy, offering improved robustness.
>
> We hope that this response can resolve your concern.
> > W1: In general, I believe this paper should lacks ablation studies.
>
> A: Additional ablation studies on various sets of a model architecture, benchmark, and test scenario can be found in the response to Reviewer M673 (ViTBase, label shifts scenario, ImageNet-C) and in Tab 15. in the Appendix (ResNet-50-BN, biased scenario, Waterbirds). Notably, the ablation study in the label shifts scenario experimentally supports our claim that PLPD is more effective than entropy. We hope that these results can resolve your concern.
>
> ### Ablation studies of DeYO on ImageNet-C under online imbalanced label shifts with ViTBase-LN
> |$S_{\theta}(x)$|$S_{\theta}(x)$|${\alpha}_{\theta}(x)$|${\alpha}_{\theta}(x)$||
> |-|-|-|-|-|
> |$Ent_{\theta}$|$PLPD_{\theta}$|$Ent_{\theta}$|$PLPD_{\theta}$|Avg Acc (%)|
> |||||52.709|
> ||||✔|54.777|
> |||✔||54.041|
> |||✔|✔|54.078|
> ||✔||| 59.975 |
> ||✔||✔|61.744|
> ||✔|✔||61.024|
> ||✔|✔|✔|61.443|
> |✔|||| 52.714 |
> |✔|||✔|56.179|
> |✔||✔||52.150|
> |✔||✔|✔|54.212|
> |✔|✔|||59.963|
> |✔|✔||✔|61.580|
> |✔|✔|✔||60.776|
> |✔|✔|✔|✔|61.304|
>
> We deeply appreciate your time and effort!
>
> The final version of the manuscript will incorporate all the responses.

---

> > ### Comment · Reviewer_NrgV · 2023-11-22
> >
> > Thanks for the efforts and explanations. I have updated my rating to 8 i..e accept.
> >
> > I really encourage the authors to incorporate all these intuitions which they have given during rebuttal in the main paper. This paper needs a lot of efforts on rewriting (including the comments from rebuttal) to make it much more impactful.
> > Again, thanks to the authors for incorporating all my comments, responding to all of them with experimental results.

---

> > > ### Author Response · Authors · 2023-11-23
> > > **Thank you for increasing your score!**
> > >
> > > We are glad to know that our response has addressed your questions.
> > >
> > > We sincerely express our gratitude for your interactive and valuable feedback. Thanks to additional discussions and experiments, we were able to effectively convey the contributions of our work.
> > >
> > > Following your comment, we have incorporated strengthened intuitions and added experimental results addressed during the rebuttal period into the revised version.
> > >
> > > Again, we would like to thank you for appreciating our work and recognizing our contributions!
> > >
> > > Best,
> > >
> > > The Authors

---

### Official Review · Reviewer_7TAy · 2023-10-30

**Soundness:** 3 good
**Presentation:** 3 good
**Contribution:** 3 good
**Rating:** 6
**Confidence:** 3

**Summary:**

The paper works on test-time adaptation with entropy minimization. While the online updates with entropy minimization can lead to error accumulation, the paper proposed to do sample selection and weighting by the proposed DeYO, which combines entropy and pseudo-label probability difference. Experiments on several datasets with various distribution shifts show the effectiveness of the method.

**Strengths:**

1. The observation and theoretical demonstration of "entropy is not enough" on the spurious correlation shifts is interesting and motivating to the method.

2. The results of the proposed method on several datasets with different distribution shifts are good, demonstrating the effectiveness of the proposed method.

**Weaknesses:**

1. The motivating observations and theoretical support are conducted on spurious correlation shifts, which is mainly on the semantic level. Can this also be found and theoretically proof for the other distribution shifts like covariate shifts?

2. In 2.3, the paper theoretically demonstrates that entropy is not enough and it is better to incorporate the CPR factors for sample selection and reweighting in test time adaptation, which is done by the proposed PLPD. However, PLPD is then combined with the common entropy method in the experiments and implementations. Then what role does entropy play in sample selection? and how it helps  PLPD to incorporate the CPR factors?

**Questions:**

1. Did the authors try some other methods like data augmentation methods to replace the transformed one? Will they also work to incorporate CPR factors?

2. How do the thresholds in eq. (9) defined?

3. What are the numbers and sizes of the patches for patch shuffling? Will these also influence the adaptation?

4. Why PLPD and entropy sample selections behave differently for different distribution shifts in Table 6? How to select different methods for different distribution shifts? Is there any theoretical support for this problem?

---

> ### Author Response · Authors · 2023-11-16
> **Response to Reviewer 7TAy (1/2)**
>
> Thank you for acknowledging that our paper is interesting and well-demonstrated both theoretically and empirically. Building on the comments you provided, we were able to further demonstrate the robustness and effectiveness of DeYO by incorporating additional theoretical explanations and experiments. We are truly grateful for the valuable comments.
>
> > W1. The motivating observations and theoretical support are conducted on spurious correlation shifts, which is mainly on the semantic level. Can this also be found and theoretically proof for the other distribution shifts like covariate shifts?
>
> A: Thank you for the constructive comments improving the scalability of our work. Firstly, we have already demonstrated it empirically as shown in Appendix G. In contrast to ImageNet-C, ImageNet-R and VISDA-2021 (Tabs. 10-13) inherently include covariate shifts. DeYO consistently exhibits significant performance on these benchmarks as well. Also, theoretically, Chunting et al. [1] demonstrated that even in situations with only covariate shifts, deep models learn spurious features present in the training data. In other words, addressing the problem of spurious correlation shifts is closely associated with alleviating covariate shifts.
>
> > W2. In 2.3, the paper theoretically demonstrates that entropy is not enough and it is better to incorporate the CPR factors for sample selection and reweighting in test time adaptation, which is done by the proposed PLPD. However, PLPD is then combined with the common entropy method in the experiments and implementations. Then what role does entropy play in sample selection? and how it helps PLPD to incorporate the CPR factors?
>
> A: In Sec. 2.3, we theoretically demonstrated that when spurious correlations exist, incorrect biases are accumulated in the model while adapting to the target domain through entropy minimization. To address this issue, we propose a new metric called PLPD. By mitigating error accumulation based on the prominent CPR factor: the shape of the object with PLPD, entropy minimization can adapt to the target domain in unlabeled situations with less bias.
>
> The impact of entropy on PLPD is as follows: even with identical PLPD values, entropy can vary depending on the sharpness of $x$'s output. Samples passing through PLPD filtering make predictions based on CPR factors, so those with sharper predictions denote a higher probability of the presence of CPR factors. Therefore, by utilizing entropy with PLPD, we can identify samples that emphasize CPR factors even within the same PLPD level.
>
> > Q1. Did the authors try some other methods like data augmentation methods to replace the transformed one? Will they also work to incorporate CPR factors?
>
> A: We compared the average PLPD measured by augmenting $x$ multiple times through AugMix [2] with the PLPD of pixel shuffling, patch shuffling, and center occlusion. AugMix employs various object-preserving augmentations such as autocontrast, equalize, rotate, and solarize, randomly applying them to create multiple augmented instances. Then, the augmented instances are mixed with the original input. AugMix is also utilized in MEMO.
>
> When using the average PLPD for a total of 8 mixed inputs in DeYO, it showed lower performance than all three object-destructive transformation methods in both ColoredMNIST and Waterbirds as shown in the tables below. The AURC, which evaluates the reliability of PLPD, also exhibited worse performance compared to the object-destructive transformations. Our proposed PLPD needs to disrupt the shape of the object to function effectively. Through the AURC values, we empirically confirmed that AugMix (object-preserving augmentation technique) cannot contribute to measuring the impact on the CPR factor.
>
> ### AugMix with PLPD
> ||ColoredMNIST||WaterBirds||
> |-|:-:|:-:|:-:|:-:|
> ||Avg Acc (%)|Worst Acc (%)|Avg Acc (%)|Worst Acc (%)|
> |DeYO with $8*x'_{AugMix}$|53.53|4.19|82.94|60.75|
> |DeYO with $x'_{pix}$|69.68|50.78|85.11|64.17|
> |DeYO with $x'_{occ}$|74.56|62.06|86.78|74.96|
> |DeYO with $x'_{pat}$|77.61|65.51|86.56|74.18|
>
> ### Evaluation of the reliability of AugMix PLPD
> ||Waterbirds|
> |-|:-:|
> ||AURC (%)
> |Entropy|56.3|
> |Pixel PLPD|20.71|
> |Patch PLPD|14.75|
> |Occ PLPD|13.77|
> |4*AugMix PLPD|35.58|

---

> ### Author Response · Authors · 2023-11-16
> **Response to Reviewer 7TAy (2/2)**
>
> > Q2. How do the thresholds in eq. (9) defined?
>
> A: We defined $\tau_{PLPD}$ based on empirical results. As shown in Fig. 7 (c), defining the threshold was not challenging. The results do not vary significantly with different thresholds, and this phenomenon remains consistent across different models, benchmarks, and test scenarios, as illustrated in Fig. 9 and Tab. 16 in Appendix I.
>
> > Q3. What are the numbers and sizes of the patches for patch shuffling? Will these also influence the adaptation?
>
> A: For the reported results, a patch size of 56x56 was employed. Consequently, a total of 16 (4x4) patches were shuffled into a 224x224 image. This information has been stated in Sec. 4.3 of the main manuscript. As demonstrated in Fig. 7 (b), unless using an excessively small patch size (similar to pixel shuffling), which could entirely distort the local features of the image, results remain similar.
>
> > Q4. Why PLPD and entropy sample selections behave differently for different distribution shifts in Table 6? How to select different methods for different distribution shifts? Is there any theoretical support for this problem?
>
> A: Tab. 6 pertains to the ablation studies on ImageNet-C. Are you inquiring about a different table? If you are referring to the ablation studies on different benchmarks between Tab. 6 in the main manuscript and Tab. 15 in Appendix, as convincingly demonstrated in Sec. 2, entropy-based sample selection is unstable in experiments under severe spurious correlations. However, fundamentally, the results from both tables are well-aligned. In other words, we believe that estimating the type or degree of distribution shift in advance is challenging. Thus, based on those results, in general, our proposed DeYO is expected to consistently deliver good performance under various circumstances.
>
> Separate from Tab. 6, we can also discuss the effectiveness of DeYO in handling different distribution shifts. Chunting et al. [1] stated that even in situations with only covariate shifts, deep models learn spurious features present in the training data, and Olivia et al. [3] identified three types of distribution shifts: 1) spurious correlation, 2) low data drift, and 3) unseen data shift, affecting generalization performance in the real world. In this regard, DeYO addresses the inherent spurious correlation problem associated with changes in the real-world input distribution, making it theoretically effective for general input distribution shifts.
>
> Also, DeYO is more robust than other baselines in wild scenarios. The reason why DeYO excels in wild scenarios is that, for the first time, it constrains the direction of adaptation through entropy minimization in distribution shift situations by focusing on the shape of the object. Even in cases where ground-truth labels exist, such as in cross-entropy learning with label shift or batch size 1, training can be disrupted by bias (label shift) or variance (batch size 1). Furthermore, in situations like TTA, error accumulation becomes more severe. Existing methods in such scenarios aim to improve performance by enhancing training stability rather than restricting the direction of adaptation. However, this only delays the worsening of severe error accumulation once it has begun and does not fundamentally reduce it.
>
> > References
> [1] Zhou, Chunting, et al. "Examining and combating spurious features under distribution shift." International Conference on Machine Learning. PMLR, 2021.
> [2] Hendrycks, Dan, et al. "AugMix: A Simple Data Processing Method to Improve Robustness and Uncertainty." International Conference on Learning Representations. 2020.
> [3] Wiles, Olivia, et al. "A Fine-Grained Analysis on Distribution Shift." *International Conference on Learning Representations*. 2022.
>
> The final version of the manuscript will incorporate all the responses.

---

> > ### Comment · Reviewer_7TAy · 2023-11-22
> > **Thanks for the response of the authors**
> >
> > The rebuttal solves most of my concerns. I would improve my score to 6.

---

> > > ### Author Response · Authors · 2023-11-23
> > > **Thank you for increasing your score!**
> > >
> > > We are glad to know that our response has addressed your questions.
> > >
> > > We sincerely express our gratitude for your valuable feedback. Thanks to additional discussions and experiments, we were able to effectively convey the contributions of our work.
> > >
> > > Again, we would like to thank you for appreciating our work and recognizing our contributions!
> > >
> > > Best,
> > >
> > > The Authors

---

> ### Author Response · Authors · 2023-11-21
> **Looking forward to your post-rebuttal feedback!**
>
> Dear Reviewer 7TAy,
>
> Thank you again for the insightful comments and suggestions! Given the limited time remaining, we eagerly anticipate your subsequent feedback. It would be our pleasure to offer more responses to further demonstrate the effectiveness of our methodology.
>
> In our previous response, we have thoroughly reviewed your comments and provided responses summarized as follows:
>
> - Explained why DeYO also addresses covariate shifts in addition to spurious correlations.
> - Explained the role of entropy on sample selection when used in conjunction with PLPD.
> - Conducted the new experiments to show the effectiveness of object-destructive transformations compared to common data augmentation methods.
> - Explained how $\tau_{PLPD}$ is selected and how many patches are used.
> - Provided more explanations related to Tab. 6 and the effectiveness of DeYO under different distribution shifts.
>
> We hope that the provided new experiments and the additional explanation have convinced you of the merits of this paper. If there are additional questions, please feel free to let us know.
>
> Additionally, we wish to express our gratitude once again to you for your insightful feedback. Incorporating your suggestions has undoubtedly enhanced the clarity and robustness of our work.
>
> We deeply appreciate your time and effort!
>
> Best regards, Authors

---

### Official Review · Reviewer_xMUg · 2023-11-03

**Soundness:** 3 good
**Presentation:** 3 good
**Contribution:** 3 good
**Rating:** 6
**Confidence:** 5

**Summary:**

The authors found that using entropy as metrics is not enough in some biased scenarios. To address this, the authors devise a new metric namely Pseudo-Label Probability Difference. The experimental results demonstrate the effectiveness of the proposed metric.

**Strengths:**

1.	The authors empirically and theoretically analysis the entropy metric for TTA.
2.	The authors devise a metric namely Pseudo-Label Probability Difference that further improves the entropy metric.
3.	The proposed method is easy to understand and implement. I believe it can be applied to real-world applications. In addition, the proposed metric only requires negligible computational cost to compute, which would not introduce obvious latency compared with EATA or SAR.

**Weaknesses:**

1.	Could the authors give simple explanation TRAP factors and CPR factors? With this, the readers may easily to capture the motivation of the proposed metric.
2.	Could the authors explain more about the motivation of the choice of patch shuffled input as x’?

**Questions:**

It would be better if the authors could explain the motivations more clearly.

---

> ### Author Response · Authors · 2023-11-16
> **Response to Reviewer xMUg (1/1)**
>
> Thank you for carefully reviewing our paper and offering a positive assessment. We appreciate your acknowledgment, particularly in relation to DeYO's exceptional performance and efficiency, highlighting its potential applicability in real-world scenarios.
>
> > W1 (Q). Could the authors give simple explanation TRAP factors and CPR factors? With this, the readers may easily capture the motivation of the proposed metric.
>
> A: Sorry for the lack of clarity in defining the terms 'TRAP' and 'CPR' factors. To enhance comprehension of these terms, we revised the introduction by incorporating examples and more explanations (Currently, the revised portions are delineated in red in the manuscript).
>
> We revised the second paragraph of page 2 in the main manuscript as follows: "Based on the observation, we introduce a theoretical proposition for identifying harmful samples, those that decrease the model's discriminability during adaptation: a sample is harmful if its prediction is more influenced by TRAin-time only Positively correlated with label (TRAP) factors (e.g., background, weather) rather than Commonly Positively-coRrelated with label (CPR) factors (e.g., structure, shape), even if its entropy is low. TRAP factors boost training performance but decrease inference performance, attributable to a discrepancy in the correlation sign with labels. If predictions mainly rely on TRAP factors, there is a high risk of wrong predictions under distribution shifts."
>
> > W2 (Q). Could the authors explain more about the motivation of the choice of patch shuffled input as x’?
>
> A: The choice of using the patch-shuffled input, denoted as $x'$, can be explained from three perspectives.
> Firstly, to capture the prominent CPR factor: the shape of the object, we had to utilize an object-destructive transformation method rather than an object-preserving transformation method such as rotation or flip. DeYo is designed to measure the difference between $x$ and $x'$, which lies in the presence or absence of the shape of the object.
> Secondly, among various object-destructive transformation methods, patch shuffling and object occlusion are options to disrupt the shape of the object in $x$. However, object occlusion presents difficulties in locating the object.
> Furthermore, unlike augmentation techniques such as CutMix [1], our work aimed to be capable of the destruction of objects using only a single image. This enables the applicability of DeYO in a batch size 1 scenario.
> Therefore, we employ patch shuffling for $x'$ on DeYO.
>
> > References
> [1] Yun, Sangdoo, et al. "Cutmix: Regularization strategy to train strong classifiers with localizable features." Proceedings of the IEEE/CVF international conference on computer vision. 2019.
>
> The final version of the manuscript will incorporate all the responses.

---

> ### Author Response · Authors · 2023-11-21
> **Looking forward to your post-rebuttal feedback!**
>
> Dear Reviewer xMUg,
>
> Thank you again for the insightful comments and suggestions! Given the limited time remaining, we eagerly anticipate your subsequent feedback. It would be our pleasure to offer more responses to further demonstrate the effectiveness of our methodology.
>
> In our previous response, we have thoroughly reviewed your comments and provided responses summarized as follows:
>
> - Revised the introduction by incorporating examples and more explanations for TRAP/CPR factors.
> - Provided three reasons why we use a patch shuffling transformation.
>
> We hope that the provided new experiments and the additional explanation have convinced you of the merits of this paper. If there are additional questions, please feel free to let us know.
>
> Additionally, we wish to express our gratitude once again to you for your insightful feedback. Incorporating your suggestions has undoubtedly enhanced the clarity and robustness of our work.
>
> We deeply appreciate your time and effort!
>
> Best regards, Authors

---

> > ### Comment · Reviewer_xMUg · 2023-11-22
> > **Thanks for authors' responses**
> >
> > I would keep my scoring.

---

> > > ### Author Response · Authors · 2023-11-23
> > > **Thanks to Reviewer xMUg**
> > >
> > > We would like to thank you again for recognizing and highly appreciating the strengths of our work.
> > >
> > > If you have any further questions or comments, please feel free to reach out to us. We will make every effort to address them.
> > >
> > > Best,
> > >
> > > The Authors

---

### Official Review · Reviewer_M673 · 2023-11-04

**Soundness:** 4 excellent
**Presentation:** 3 good
**Contribution:** 3 good
**Rating:** 8
**Confidence:** 4

**Summary:**

This paper presents a new method for test-time adaptation (TTA) called Destroy Your Object (DeYO) that uses a novel confidence metric called Pseudo-Label Probability Difference (PLPD) to improve the adaptation performance and stability of test-time adaptation methods. The authors demonstrate the limitations of entropy as a confidence metric and compare the performance of DeYO with other TTA methods on the ImageNet-C and ImageNet-R benchmarks under various mild and wild test scenarios.

**Strengths:**

The proposed DeYO method is simple yet effective for improving the stability and performance of entropy-based TTA.

The idea and motivation behind Pseudo-Label Probability Difference (PLPD) are novel and interesting, providing new insights for the community.

The paper is strong on the empirical side. Extensive experiments with various model architectures, datasets, and mild/wild test scenarios are thorough.

**Weaknesses:**

The proposed terms “TRAP” and “CRP” are a bit hard to understand. The authors could refine the name and give more high-level/easy-understanding explanations about them in the Introduction.

For parameter sensitivity analyses in Figure 7, could the authors report more results under different model architectures, datasets and test scenarios? This helps demonstrate the hyperparameters’ generality.

Ablation studies in Table 6 are also highly encouraged to be conducted on different models, datasets and test scenarios.

**Questions:**

Pls refer to Weakness.

---

> ### Author Response · Authors · 2023-11-16
> **Response to Reviewer M673 (1/2)**
>
> Thank you for positively evaluating our proposed method, DeYO. We appreciate your recognition of the novelty in our idea and motivation, acknowledging its contribution to providing new insights to the community. Moreover, we are grateful for your acknowledgment of the strength of DeYO on the empirical side.
>
> Based on your comments, we revised the main manuscript to improve the readability of the manuscript and conducted experiments to further prove the robustness of DeYO.
>
> > W1. The proposed terms “TRAP” and “CPR” are a bit hard to understand. The authors could refine the name and give more high-level/easy-understanding explanations about them in the Introduction.
>
> A: Sorry for the lack of clarity in defining the terms 'TRAP' and 'CPR' in the introduction. To enhance comprehension of these terms, we revised the introduction by incorporating examples and more explanations (Currently, the revised portions are delineated in red in the manuscript).
>
> We revised the second paragraph of page 2 in the main manuscript as follows: "Based on the observation, we introduce a theoretical proposition for identifying harmful samples, those that decrease the model's discriminability during adaptation: a sample is harmful if its prediction is more influenced by TRAin-time only Positively correlated with label (TRAP) factors (e.g., background, weather) rather than Commonly Positively-coRrelated with label (CPR) factors (e.g., structure, shape), even if its entropy is low. TRAP factors boost training performance but decrease inference performance, attributable to a discrepancy in the correlation sign with labels. If predictions mainly rely on TRAP factors, there is a high risk of wrong predictions under distribution shifts."
>
> We have kept the term as it is, as we could not think of a good alternative. However, if it still remains unclear in the current revised manuscript, we will modify the term as well. Please feel free to share your thoughts.
>
> > W2. For parameter sensitivity analyses in Fig. 7, could the authors report more results under different model architectures, datasets and test scenarios? This helps demonstrate the hyperparameters’ generality.
>
> A: Thank you for the suggestion. Following your comment, we experimented with the parameter sensitivity of DeYO under a different model architecture (ViTBase), dataset (WaterBirds), and test scenario (wild scenario: online imbalanced label shifts and biased scenario: spurious correlation). We still observed consistent results with those of Fig. 7 (ResNet-50-BN, ImageNet-C, mild scenario) as shown in the tables:
> ### Augmentation Type
> |       | ImageNet-C (ViTBase-LN, wild scenario: label shifts) | WaterBirds (ResNet-50-BN, biased scenario) |  WaterBirds (ResNet-50-BN, biased scenario) |
> |-------|:----------------------------------------------------:|:------------------------------------------:|:-------------:|
> |       |                      Avg Acc (%)                     |                 Avg Acc (%)                | Worst Acc (%) |
> | Pixel |                         52.01                        |                    85.84                   |     65.42     |
> | Patch |                         61.30                        |                    88.16                   |     74.18     |
> | Occ   |                            60.24                         |                    87.69                   |     73.55     |
> ### Number of Patches
> |     | ImageNet-C (ViTBase-LN, wild scenario: label shifts) | WaterBirds (ResNet-50-BN, biased scenario) | WaterBirds (ResNet-50-BN, biased scenario) |
> |-----|:----------------------------------------------------:|:------------------------------------------:|:------------------------------------------:|
> |     |                      Avg Acc (%)                     |                 Avg Acc (%)                |                Worst Acc (%)               |
> | 2X2 |                         62.47                        |                    86.98                   |                    74.96                   |
> | 3X3 |                         60.73                        |                    87.26                   |                    75.43                   |
> | 4X4 |                         61.30                        |                    88.16                   |                    74.18                   |
> | 5X5 |                         59.46                        |                    87.12                   |                    72.14                   |
> | 6X6 |                         61.04                        |                    87.40                   |                    72.14                   |

---

> ### Author Response · Authors · 2023-11-16
> **Response to Reviewer M673 (2/2)**
>
> ### $\tau_{PLPD}$
> |     | ImageNet-C (ViTBase-LN, wild scenario: label shifts) | WaterBirds (ResNet-50-BN, biased scenario) | WaterBirds (ResNet-50-BN, biased scenario) |
> |-----|:----------------------------------------------------:|:------------------------------------------:|:------------------------------------------:|
> |     |                      Avg Acc (%)                     |                 Avg Acc (%)                |                Worst Acc (%)               |
> | 0.0 |                         60.86                        |                    84.54                   |                    58.84                   |
> | 0.2 |                         61.30                        |                    87.31                   |                    71.67                   |
> | 0.3 |                         61.45                        |                    87.72                   |                    72.93                   |
> | 0.4 |                         61.25                        |                    87.45                   |                    71.67                   |
> | 0.5 |                         60.26                        |                    88.16                   |                    74.18                   |
> | 0.6 |                           -                          |                    88.06                   |                    74.80                   |
> | 0.7 |                           -                          |                    87.59                   |                    74.02                   |
>
> For Waterbirds, it involves a 2-class classification task. Therefore, the value of $\tau_{PLPD}$ should be larger than that of ImageNet-C, which has 1000 classes, to adequately capture the changes induced by transformations.
>
> > W3. Ablation studies in Tab. 6 are also highly encouraged to be conducted on different models, datasets and test scenarios.
>
> A: The ablation studies on WaterBirds (a different dataset and biased scenario: spurious correlation) have been provided in Appendix H. Additionally, based on your comment, we validated DeYO under a different model architecture (ViTBase), and test scenario (wild scenario: online imbalanced label shifts). We confirmed that the ablation results of ViTBase-LN under a wild scenario show a consistent trend with those of ResNet-50-BN under a biased scenario and ResNet-50-BN under a mild scenario (Tab. 6 in the main manuscript).
>
> ### Ablation studies of DeYO on ImageNet-C under online imbalanced label shifts with ViTBase-LN
> | $S_{\theta}(x)$ | $S_{\theta}(x)$ | ${\alpha}_{\theta}(x)$ | ${\alpha}_{\theta}(x)$ |  |
> |:---:|:---:|:---:|:---:|:---:|
> | $Ent_{\theta}$ | $PLPD_{\theta}$ | $Ent_{\theta}$ | $PLPD_{\theta}$ | Avg Acc (%) |
> |  |  |  |  | 52.709 |
> |  |  |  | &#10004; | 54.777 |
> |  |  | &#10004; |  | 54.041 |
> |  |  | &#10004; | &#10004; | 54.078 |
> |  | &#10004; |  |  | 59.975 |
> |  | &#10004; |  | &#10004; | 61.744 |
> |  | &#10004; | &#10004; |  | 61.024 |
> |  | &#10004; | &#10004; | &#10004; | 61.443 |
> | &#10004; |  |  |  | 52.714 |
> | &#10004; |  |  | &#10004; | 56.179 |
> | &#10004; |  | &#10004; |  | 52.150 |
> | &#10004; |  | &#10004; | &#10004; | 54.212 |
> | &#10004; | &#10004; |  |  | 59.963 |
> | &#10004; | &#10004; |  | &#10004; | 61.580 |
> | &#10004; | &#10004; | &#10004; |  | 60.776 |
> | &#10004; | &#10004; | &#10004; | &#10004; | 61.304 |
>
> The final version of the manuscript will incorporate all the responses.

---

> ### Author Response · Authors · 2023-11-21
> **Looking forward to your post-rebuttal feedback!**
>
> Dear Reviewer M673,
>
> Thank you again for the insightful comments and suggestions! Given the limited time remaining, we eagerly anticipate your subsequent feedback. It would be our pleasure to offer more responses to further demonstrate the effectiveness of our methodology.
>
> In our previous response, we have thoroughly reviewed your comments and provided responses summarized as follows:
>
> - Revised the introduction by incorporating examples and more explanations for TRAP/CPR factors.
> - Conducted the new experiments to show DeYO is not sensitive to hyperparameters regardless of various architectures, scenarios, and benchmarks.
> - Conducted the new experiments to show DeYO has a consistent trend regardless of various architectures, scenarios, and benchmarks.
>
> We hope that the provided new experiments and additional explanation have convinced you of the merits of this paper. If there are additional questions, please feel free to let us know.
>
> Additionally, we wish to express our gratitude once again to you for your insightful feedback. Incorporating your suggestions has undoubtedly enhanced the clarity and robustness of our work.
>
> We deeply appreciate your time and effort!
>
> Best regards, Authors

---

> > ### Author Response · Authors · 2023-11-23
> > **Thanks to Reviewer M673**
> >
> > We would like to thank you again for recognizing and highly appreciating the strengths of our work.
> >
> > If you have any further questions or comments, please feel free to reach out to us. We will make every effort to address them.
> >
> > Best,
> >
> > The Authors

---

### Author Response · Authors · 2023-11-21
**General Response**

We sincerely appreciate the reviewers' time and invaluable feedback. Before addressing each reviewer's comments, we would like to express our gratitude to the reviewers for their recognition of our research. Below is a summary highlighting the strengths of our work.

### **Our novelty and contributions recognized by reviewers**

- The problem is **well motivated** and **the** **empirical observation and theoretical demonstration of "entropy is not enough"** on the spurious correlation shifts is **interesting,** which has never been explored in test-time adaptation literature. [M673, xMUg, 7ATy, NrgV]
- The idea and motivation behind Pseudo-Label Probability Difference (PLPD) are **novel** and **interesting, providing new insights for the community**. [M673, xMUg]
- The proposed method is **easy to understand and implement**. [M673, xMUg, 7TAy]
- The proposed metric **only requires negligible computational cost** to compute, which would not introduce obvious latency compared with EATA or SAR. It can be **applied to real-world applications**. [xMUg]
- The paper is **strong on the empirical side**. Extensive experiments with **various model architectures, datasets, and mild/biased/wild test scenarios are thorough**, demonstrating the effectiveness of the proposed method. [M673, xMUg, 7TAy]

We respond to each reviewer's comments in detail below. In accordance with the reviewers' suggestions, we revised the paper and uploaded the revised version, which we believe enhanced the paper's strength. Here, we would like to highlight a major revision or update:

### **Introduction**

- We revised the introduction by incorporating examples and more explanations for TRAP/CPR factors suggested by Reviewers M673 and xMUg: Section 1.

### **Related Work**

- We discussed more papers and related topics including SENTRY [1], and Self-training Avoids Using Spurious Features Under Domain Shift [2] suggested by Reviewer NrgV: Appendix D

### **Experiments**

- Additional experiments on hyperparameter sensitivity & ablation studies of DeYO suggested by Reviewer M673
- Experiments (including MEMO [3] and SENTRY [1]) to show the effectiveness of object-destructive transformations compared to common data augmentation methods suggested by Reviewers 7TAy and NrgV
- Experiment to show the effectiveness of sample selection based on PLPD during pretraining suggested by Reviewer NrgV
- Experiments to show that multiple object-preserving augmentations are not suitable for PLPD suggested by Reviewers 7TAy and NrgV.
- We **added all the additional experimental results** to the revised version of our paper.

### **Discussion**

- Additional discussion on how PLPD makes models more robust in wild scenarios compared to entropy suggested by Reviewer NrgV.
- Additional discussion on whether PLPD contributes to robust adaptation in covariate shifts, not just in spurious correlation shifts suggested by Reviewer 7TAy.

We will also incorporate the responses during a discussion phase into the final manuscript.

> References
[1] Prabhu, Viraj, et al. "Sentry: Selective entropy optimization via committee consistency for unsupervised domain adaptation." International Conference on Computer Vision. 2021.
[2] Chen, Yining, et al. "Self-training avoids using spurious features under domain shift." Neural Information Processing Systems. 2020.
[3] Zhang, Marvin, et al. "Memo: Test time robustness via adaptation and augmentation." Neural Information Processing Systems. 2022.

---

### Meta-Review · Area_Chair_GuVz · 2023-12-05

**Metareview:**

This paper proposes a new approach to test-time adaptation (TTA). Entropy minimization of the test-samples is a well-known method for TTA used in methods like TENT. The paper shows that under certain conditions, it may not be suitable. It proposes a method named "Destroy Your Object" (DeYO) based on a new confidence metric termed as Pseudo-Label Probability Difference (PLPD), which measures how much the shape of an object influences the prediction. It is essentially defined as the difference between the predictions before and after applying an object-destructive transformation.

The proposed idea is novel and yet simple and effective, and performs well in practice. The paper does well both in terms of empirical evaluation and theoretical analysis, and all the reviewers rated the paper favorably. Some of the concerns and questions raised by the reviewers were addressed extensively which the reviewers appreciated a lot.

Based on the reviews, the authors' response, and my own reading of the work, I believe this work would be a nice addition to the topic of TTA and recommend acceptance.

**Justification For Why Not Higher Score:**

I think spotlight would be an appropriate recommendation; the paper definitely should be given attention given that it sheds light on a well-known method regarding its limitations and proposed a new method. However, it need not have a full oral presentation.

**Justification For Why Not Lower Score:**

The paper has several strong points, as mentioned in the meta-review, which make it suitable for spotlight presentation.

---

### Decision · Program_Chairs · 2024-01-16

Accept (spotlight)